# ITERATIVELY LEARNING FROM THE BEST

## ABSTRACT

We study a simple generic framework to address the issue of bad training data; both bad labels in supervised problems, and bad samples in unsupervised ones. Our approach starts by fitting a model to the whole training dataset, but then iteratively improves it by alternating between *(a)* revisiting the training data to select samples with lowest *current* loss, and *(b)* re-training the model on *only* these selected samples. It can be applied to any existing model training setting which provides a loss measure for samples, and a way to refit on new ones. We show the merit of this approach in both theory and practice. We first prove statistical consistency, and linear convergence to the ground truth and global optimum, for two simpler model settings: mixed linear regression, and gaussian mixture models. We then demonstrate its success empirically in *(a)* saving the accuracy of existing deep image classifiers when there are errors in the labels of training images, and *(b)* improving the quality of samples generated by existing DC-GAN models, when it is given training data that contains a fraction of the images from a different and unintended dataset. The experimental results show significant improvement over the baseline methods that ignore the existence of bad labels/samples.

## 1 INTRODUCTION

This paper is **motivated** by the problem that training large machine learning models requires well-curated training sets with lots of samples. This is an issue in both supervised and unsupervised settings. For *supervised* problems (e.g. multi-label classification) we need very many properly labeled examples; generating these requires a lot of human effort. Errors in labels can significantly affect the accuracy of naive training. For example, a faulty CIFAR-10 dataset with 20% of automobile images mis-labeled as "airplane" (and so on for the other classes) leads to the accuracy of a neural architecture like WRN of Zagoruyko & Komodakis (2016) to go from over 90% to about 70%. Even in *unsupervised* problems, where there are no labels, spurious samples are a nuisance. Consider for example the task of training a generative adversarial network (GAN) Goodfellow et al. (2014) to produce realistic images of faces. While this has been shown to work well using a dataset of face images [1], like Celeb-A Liu et al. (2015), it degrades quickly if the training dataset is corrupted with non-face images. Our experiments illustrate these issues.

**Main idea**    We study a simple framework that addresses the problem of spurious data in both supervised and unsupervised settings. Note that if we knew which samples were clean, we could learn a model; conversely, if we had a good model, we could find the clean samples. But we do not have either; instead we have a chicken-and-egg problem: finding the best subset of samples needs a model, but finding a model needs a subset of samples. In this paper, we take the natural approach to this problem by *(0)* fitting an initial model on all samples, and then alternating between *(1)* finding the best subset of samples given a model, and *(2)* fitting a model given a subset. [2] Our approach is most related to the works of Bhatia et al. (2015) and Vainsencher et al. (2017), and we contrast to these in the related work section.

---

[1] https://github.com/carpedm20/DCGAN-tensorflow

[2] Our framework sounds initially similar to EM-style algorithms like $k$-means. Note however that EM needs to postulate a model for all the data points, while we search over a subset and do not worry about the loss on corrupted points. We are alternating between a simple search over subsets and a fitting problem on only the *selected* subset; this is not an instance of EM.

Our approach can be applied to any existing model that provides (a) confidence scores for samples, and (b) a way to train the model for a set of samples. Indeed in our experiments we use existing models and training code, but with iterative (re)selections of samples as above. In supervised settings, any model that maps features to responses typically associates a confidence to every sample, e.g., in linear regression, given a model $\theta$, the confidence in a sample $(\mathbf{x}_i, y_i)$ increases as its squared error $(y_i - \mathbf{x}_i^\top \theta)^2$ decreases. For a neural network classifier, on the other hand, the confidence can be measured by the cross entropy loss between the output probability distribution and the true distribution. In unsupervised settings, the confidence of a sample is given by its "likelihood" under the model. If we are lucky, e.g., in Gaussian mixture models, this likelihood is explicit and can be evaluated. However, if there is no explicit likelihood, we need to rely on a surrogate. E.g., in our GAN setting below, we rely on the output of the discriminator as a confidence measure. We also note an important **caveat:** this paper is not focused on (and may not be effective for) the settings of adversarial examples, or of gross corruptions of features. Rather, our focus is to address the issue of bad labeling or curation of data, as motivated above.

### OUR RESULTS AND PAPER OUTLINE

We show that our framework (outlined in Section 3) can address the spurious sample problem, both theoretically and empirically and for both supervised and unsupervised problems:

**(1) Mixed linear regression (Section 4) and Gaussian mixture models (Section 5)**: For each of these mixture problem settings, under the standard statistical assumptions, we establish both statistical consistency and linear convergence of our iteratively updated model parameters. This holds true from any initial point; the convergence is to the mixture component that is closest to the initial point.

**(2) Neural network classifiers (Section 6) :** We investigate image classification in CIFAR-100, CIFAR-10 and MNIST datasets (each with a different and appropriate *existing* neural architecture) in the setting where there are *systematic* errors in the labels – i.e. every bad example in one label class comes from the same other label class. We show that our framework can significantly improve the classification accuracy over the baseline method that ignores the existence of bad samples.

**(3) Deep generative models (Section 7):** We show both quantitatively and qualitatively the improvement of using our algorithm for image generation when the dataset consists of two types of images. Specifically, investigate training a GAN to *(a)* generate face images as in the "intended" Celeb-A dataset, but when the given training data has a fraction of the samples being the "spurious" CIFAR-10 images, and *(b)* generate intended MNIST digits when training data contains a fraction of samples from the spurious Fashion-MNIST dataset. A *crucial innovation* here is that the output of the discriminator network in the GAN can be used as a confidence metric for the training samples.

In Section 8, we discuss the positives and negatives of our results, underlying intuition and also some future directions. All of our experiments were done on clearly identified publicly available neural network architectures, datasets, and training software. We also clearly describe our error and noise effects; we expect them to be easily reproducible.

## 2 RELATED WORK

**Alternating minimization for robustness**    Two works are most related to ours. Bhatia et al. (2015) proposes an iterative hard thresholding algorithm for adversarial noise in linear regression responses; they establish statistical guarantees and linear convergence under this harder noise model. We focus on multiple mixed linear regression, a different setting that results in an easier noise model, and we provide convergence guarantee for recovering any single mixture component starting from a local region. In addition we also prove these for Gaussian mixture models, which is not studied in their work. Vainsencher et al. (2017) proposes what is essentially a soft version of our method, and proves local convergence and generalization. However they do not have any initialization, and hence no global or consistency guarantees. Neither of these empirically explore the overall approach for more complex / neural network models.

**Robust regression** Recent work on robust regression consider strong robustness in terms of both the inputs and the outputs and provide guarantees for constant corruption ratio Diakonikolas et al. (2018); Prasad et al. (2018); Klivans et al. (2018). Chen et al. (2013); Balakrishnan et al. (2017a); Liu et al. (2018) study the high dimensional setting under this regime. These algorithms usually require much more computation, compared with methods for dealing with noisy outputs, e.g., Bhatia et al. (2015). Another type of robustness considers heteroscedastic noise for every sample. From the learning perspective, Anava & Mannor (2016); Chaudhuri et al. (2017) both require strong constraints on the noise model which largely reduces the degree of freedom. Anava & Mannor (2016) considers the online learning setting, while Chaudhuri et al. (2017) considers the active learning setting.

**Mixed linear regression** Alternating minimization type algorithms are used for mixed linear regression with convergence guarantee Yi et al. (2014). Chen et al. (2014) provides a convex formulation for the problem. Balakrishnan et al. (2017b) shows expectation-maximization (EM) algorithm is provably effective for a set of tasks including the basic setting of mixed linear regression and Gaussian mixture model. Sedghi et al. (2016) extends to the multiple component setting, where provable tensor decomposition methods are used. More recently, Li & Liang (2018) gives a more general result for learning mixtures of linear regression. On the other hand, Ray et al. (2018) studies the problem of finding a single component in mixed linear regression problem using side information.

**Noisy labels** Classification tasks with noisy labels are also of wide interest. Frénay et al. (2014) gives an overview of the related methods. Theoretical guarantee for noisy binary classification has been studied under different settings Scott et al. (2013); Natarajan et al. (2013); Menon et al. (2016). More recently, noisy label problems have been studied for deep neural network based models. Reed et al. (2014) and Malach & Shalev-Shwartz (2017) develop the idea of bootstrapping and query-by-committee into neural network based models. On the other hand, Khetan et al. (2018) and Zhang & Sabuncu (2018) provide new losses for training under the noise. Sukhbaatar & Fergus (2014) adds a noise layer into the training process, while Ren et al. (2018b) provides a meta-algorithm for learning the weights of all samples by referencing to a clean subset of validation data during training. Ren et al. (2018a) consider building robust classifiers for the positive-unlabeled classification problem using $l_0$ norm penalization.

**Others** Bora et al. (2018) studies the problem of noisy images for GANs. In their setting, all images are of low quality under some measurements, while in our problem, we consider the image dataset consists of (possibly more than) two types of images, which is different. There are also studies that consider noisy samples more generally. The classical RANSAC method Fischler & Bolles (1981) provides a paradigm for fitting a model to experimental data, and can be potentially used for the noisy data setting. However, the algorithm requires multiple times of random sampling to find a valid consensus score, which is conceptually different from our idea of doing iterative filtering. EM algorithm Moon (1996) can also be used for detecting noisy samples, e.g., in the setting of mixture models Balakrishnan et al. (2017b). However, our algorithm does not need to know the likelihood for the bad samples for the iterative updates.

## 3 ITERATIVELY LEARNING FROM THE BEST ( ILFB )

We now describe the setup and algorithm, in a unified way that covers both supervised and unsupervised settings. We will then specialize to the many specific problems we study. We are provided with *samples* $s_1, \cdots, s_n$, using which we want to train a *model* parameterized by $\theta \in \mathbb{R}^d$. We are also provided with a loss function; let $f_\theta(s)$ denote the loss of a sample $s$ when the parameters are $\theta$. Let $\tau \in [0, 1]$ be the fraction of data we want to fit. With this setup, the idea of learning from the best can be viewed as trying to solve

$$\min_\theta \min_{S:|S|=\tau n} \sum_{i \in S} f_\theta(s_i).$$

We do this by alternating between finding $S$ and finding $\theta$, as described below in Alg. 1.

---

**Algorithm 1** ILFB Update

---

1: **input:** samples $\{s_i\}_{i=1}^n$, fraction $\tau$, max. iteration number $T$
2: **(optional) initialize:** set $t \leftarrow 0$ and initially fit all samples $\theta_0 \leftarrow \arg\min_\theta \sum_{i\in[n]} f_\theta(s_i)$,
3: **while** $t < T$ **do**
4:     calculate current losses $f_{\theta_t}(s_i)$ for all samples $s_i$, $i \in [n]$
5:     find new best set of samples

$$S_t \leftarrow \arg\min_{S:|S|=\tau n} \sum_{i\in S} f_{\theta_t}(s_i)$$

    by sorting them according to losses $f_{\theta_t}(s_i)$ and choosing the ones with smallest losses.
6:     fit a model for this new set

$$\theta_{t+1} \leftarrow \arg\min_\theta \sum_{i\in S_t} f_\theta(s_i)$$

7:     $t \leftarrow t + 1$
8: **return:** $\theta_T$

---

**Comments**    Steps 4, 5 and 6 above can be done approximately, when exact solutions are unavailable. For example, in our experiments involving deep neural networks (both the classifiers and the generative models), for step 6 we do not fit a model exactly, but rather do training using stochastic gradient descent. Also, for our experiments on training GANs on the best subset, we do not have an explicit loss; we instead use the loss at the output of the discriminator as a surrogate (more details on that below). Similarly, the initialization step above can possibly be replaced with some problem-setting dependent alternative, if one is available.

In the following, we describe how this directly specializes to each of our four settings.

## 4    ILFB for Mixed Linear Regression

We first describe ILFB as specialized to the case when the $\theta$ represents the parameters of a linear regression, and then describe the statistical setting of mixed linear regression where we provide rigorous statistical guarantees on its performance.

**Algorithm**    The samples are $s_i = (\mathbf{x}_i, y_i)$ with features $\mathbf{x}_i \in \mathbb{R}^d$ and responses $y_i \in \mathbb{R}$, and $\theta \in \mathbb{R}^d$ are the parameters of the linear model. The loss is $f_\theta(\mathbf{x}, y) = (y - \mathbf{x}^\top \theta)^2$. The algorithm thus initializes $\theta_0$ by doing an ordinary least squares (OLS) on all the samples, and then alternates between finding the set of $\tau n$ samples with the lowest squared error for the current $\theta$, and then finding the new $\theta$ by doing an OLS on this set.

**Statistical Setting**    In the standard and widely studied mixed linear regression problem there are $n$ samples $(\mathbf{x}_i, y_i)$, with features $\mathbf{x}_i \sim \mathcal{N}(0, I_d)$ being standard gaussians in $d$ dimensions. The full set of samples $S = [n]$ can be splitted into $m$ sets $S = \cup_{j\in[m]} S^j$, each corresponding to samples that generate from one mixture component, $|S^j| = \tau_j^\star n$. The response variables $y_i$ are given by:

$$y_i = x_i^\top \theta^{(j)} + \epsilon_i, \text{ for } i \in S^j \tag{1}$$

where $\epsilon_i \sim \mathcal{N}(0, \sigma^2)$ is the additive noise, and $\theta^{(j)}$s are assumed to be unit norm vectors. We analyze our algorithm in the fresh sample setting, where iteration $t$ finds $\theta_{t+1}$ from $\theta_t$ using a new set of samples. This too is standard in the statistical analysis of mixed linear regression, and corresponds to the case of using mini-batches of samples to learn a model.

We now prove two results: the first shows that the iterates converge linearly to some $\theta^{(j)}$ once $\theta_t$ is closer enough to $\theta^{(j)}$ than any of the other components, and the second shows that the simple initialization step ensures a $\theta_0$ that is close enough to the dominant component $\theta^{(j)}$ under mild assumptions.

**Theorem 1** (**local linear convergence**). *For the mixed linear regression setting as above, we run ILFB as shown in Algorithm 1. Suppose for some $j \in [m]$, the iterate $\theta_t$ and $\tau$ satisfy:*

*(C1)* $\tau \leq 0.99\tau_j^\star$;

*(C2)* $\|\theta_t - \theta^{(j)}\|_2 \leq C(\tau)\min_{l \in [m] \setminus \{j\}} \|\theta_t - \theta^{(l)}\|_2 - \sqrt{1 - C^2(\tau)}\sigma$, *where* $C(\tau) = \min\{\frac{c_1 c_3 \tau}{1-\tau}, 1\}$.

*Then, the next iterate $\theta_{t+1}$ of the algorithm satisfies*

$$\|\theta_{t+1} - \theta^{(j)}\|_2 \leq c_1 \|\theta_t - \theta^{(j)}\|_2 + (c_1 + c_2)\sigma$$

*with high probability (i.e., with probability $1 - n^{-c_0}$), provided the number of samples $n \geq c\frac{d \log d}{c_2^2 \tau}$, where $c_1 \in (0, 1), c, c_0, c_2, c_3$ are constants.*

Theorem 1 establishes a linear dependence of the sample complexity on the dimension $d$. The required sample size increases as $\tau$ gets smaller. For $\tau$ large enough such that $C(\tau) = 1$, the condition on $\theta_t$ does not depend on $\sigma$. However, notice that for smaller $\tau$, the condition of $\theta_t$ tolerates smaller noise. This is because, even if $\theta_t$ is very close to $\theta^{(j)}$, when the noise and the density of samples from other mixture components are both high, the number of samples from other components selected by the current $\theta_t$ would still be quite large, and the update will not converge to $\theta^{(j)}$.

The proof for the theorem is in Appendix C.1, and synthetic experiments verifying the performance of ILFB are in Appendix C.2. In Appendix C.2, we show that our guarantee for the algorithm is tight, in the sense that there always exist a $\sigma$ dependency even if we take infinite samples. In the noiselss setting, our theorem guarantees local linear convergence with exact recovery.

## 5 ILFB FOR GAUSSIAN MIXTURE MODEL

In this section, we describe ILFB applied to the setting of Gaussian mixture model, and provide statistical guarantees on its performance.

**Algorithm**    The samples $s_i = \mathbf{x}_i$ are points in $\mathbb{R}^d$, $\theta \in \mathbb{R}^d$ is the mean we want to estimate, and $f_\theta(\mathbf{x})$ is proportional to the negative log-likelihood, i.e., $f_\theta(\mathbf{x}) = \|\mathbf{x} - \theta\|_2^2$. The algorithm thus initializes the mean by finding the average of all $n$ points, and then iteratively finds the set of $\tau n$ points closest to the current mean, and updating the mean to be the average of this set.

**Statistical setting**    Given a dataset $\mathcal{D} = \{\mathbf{x}_i \mid i \in S = \{1, \cdots, n\}\}$, where $S$ can be splitted into disjoint sets $S = \cup_{j \in [m]} S^j$, $|S^j| = \tau_j^\star n$. The samples are generated following:

$$\mathbf{x}_i \sim \mathcal{N}\left(\theta^{(j)}, \sigma^2 I\right), \text{ for } i \in S^j. \tag{2}$$

We now prove two results: the first shows that iterates converge linearly under the fresh sample setting in a local region, while the second shows with some additional conditions, that the initialization of ILFB falls into the local region of the dominant component.

**Theorem 2** (**local linear convergence**). *We run Alg. 1 on the given dataset $\mathcal{D}$ generated following (2). Suppose for some $j \in [m]$, the iterate $\theta_t$ and parameter $\tau$ in Algorithm 1 satisfy:*

*(C1)* $\tau \leq 0.99\tau_j^\star$;

*(C2)* $\|\theta_t - \theta^{(j)}\|_2 \leq 0.9 \min_{l \in [m] \setminus \{j\}} \|\theta_t - \theta^{(l)}\|_2 - c_3\sigma\sqrt{\log \frac{2(1-\tau)}{\tau}}\mathbf{1}_{\{\tau < \frac{1}{2}\}}$

*then, with high probability (i.e., with probability $1 - n^{-c_0}$),*

$$\|\theta_{t+1} - \theta^{(j)}\|_2 \leq c_1 \|\theta_t - \theta^{(j)}\|_2 + c_2\sqrt{d}\sigma,$$

*where $c_1 \in (0, 1), c_0, c_2, c_3$ are constants.*

Table 1: **Classification with systematic label error:** Performance for MNIST, CIFAR-10, CIFAR-100 datasets as the ratio of good samples varies from 60% to 90%. Here **Baseline** : Naive training using all the samples; **ILFB** : Our iterative update algorithm; **Oracle** : Training with all good samples. We see significant improvement of **ILFB** over **Baseline** for all the settings.

| dataset | MNIST | | | CIFAR-10 | | | CIFAR-100 | | |
|---|---|---|---|---|---|---|---|---|---|
| $\frac{\text{# good}}{\text{# total}}$ | **Baseline** | **ILFB** | **Oracle** | **Baseline** | **ILFB** | **Oracle** | **Baseline** | **ILFB** | **Oracle** |
| 60% | 70.26 | 90.00 | 94.30 | 64.63 | 75.16 | 91.26 | 58.28 | 65.25 | 80.39 |
| 70% | 85.95 | 92.09 | 94.50 | 71.05 | 86.09 | 92.25 | 66.75 | 73.83 | 81.65 |
| 80% | 92.62 | 94.30 | 94.61 | 78.84 | 89.14 | 92.85 | 72.68 | 78.70 | 82.50 |
| 90% | 93.88 | 94.41 | 94.92 | 84.12 | 91.03 | 93.47 | 78.12 | 81.10 | 83.41 |

Here, the 0.9 factor in the assumption is chosen for convenience of the analysis. Essentially, what we need is this value being less than one. For $\tau < \frac{1}{2}$, we require an additional separation term that depends on $\sigma$. This term essentially implies a $\sqrt{\log k}\sigma$ separation distance (consider all mixture components have weights $1/k$), which is the lower bound for computationally efficient algorithm to learn a Gaussian mixture model according to Regev & Vijayaraghavan (2017). Intuitively, given the required initialization condition, the samples selected by the initialization parameter contain more than half of the good samples, and we can show that the algorithm is guaranteed to converge. Notice that there is no dependency between $n$ and $d$. In the extreme case where $\sigma = 0$, we only need $n$ to be a constant value: from any good initialization point, the algorithm will converge to $\theta^{(j)}$ in one step.

The proof for the theorem is in Appendix D.1, and synthetic experiments verifying the performance of ILFB are in Appendix D.2. Again, in Appendix D.2, we show that the our guarantee of the algorithm is tight.

# 6 ILFB FOR DEEP IMAGE CLASSIFICATION WITH SYSTEMATIC TRAINING LABEL ERROR

**Experimental setting** We consider *systematic errors in labels*, i.e. the setting where all the samples that actually come from a class "a" are given the *same* bad label "b", a setting that is less benign than one with more randomness in the errors. We investigated the ability of our framework to account for these errors for the following dataset – neural architecture pairs:
*(a)* MNIST LeCun et al. (1998) dataset with a standard 2-layer CNN [3]
*(b)* CIFAR-10 Krizhevsky & Hinton (2009) with a 16-layer WideResNet [4] Zagoruyko & Komodakis (2016)
*(c)* CIFAR-100 [5] with a 28-layer WideResNet [6]
In each of these cases, the neural network has excellent performance when there are no label errors in training; this degrades significantly as the fraction of spurious samples increases.

**Algorithm** In each case, the algorithm involves first training the NN on all the samples, and then iteratively alternating between *(a)* picking the fraction $\tau$ of samples in each label class with the lowest cross-entropy loss under the current $\theta$, and *(b)* retraining the model from scratch with these picked samples, to get a new $\theta$. Training was done on MXNet. Mapping back to the development in Section 3, here $\theta$ represents the weights of the neural network, and $f_\theta(s)$ is the standard cross-entropy loss used in training.

**Further details** When training on all the samples, we run simple stochastic gradient descent algorithm with initial learning rate 0.5/0.1/0.1 and batch size 1000/256/64 for MNIST/CIFAR-10/CIFAR-100, respectively. The learning rate is divided by 5 at the 50-th epoch and each ex-

---

[3]follow the implementation in https://mxnet.incubator.apache.org/tutorials/python/mnist.html

[4]cifar_wideresnet16_10 from the online model zoo: https://gluon-cv.mxnet.io/api/model_zoo.html

[5]We use the coarse labels as the classification target

[6]WideResNet: depth 28, width factor 10, dropout rate 0.3

Table 2: **Generative models from mixed training data:** A **quantitative** measure of the efficacy of our approach is to find how many of the good training samples the final discriminator can identify; this is shown here for the three different "good"/"bad" dataset pairs. For each pair, the fraction of "good" samples is 90%, 80% or 70%. The table depicts the ratio of the good samples in the training data that are *recovered* by the discriminator when it is run on the training samples. The higher this fraction, the more effective the generator. For MNIST-Fashion and CelebA-CIFAR10, our approach shows significant improvements with iteration count. For CIFAR10-CelebA dataset, the error is extremely simple to be corrected, likely because faces are easier to discriminate against when compared to natural images.

|  | MNIST(good)-Fashion(bad) | | | CIFAR10(good)-CelebA(bad) | | | CelebA(good)-CIFAR10(bad) | | |
| --- | --- | --- | --- | --- | --- | --- | --- | --- | --- |
| orig | 90% | 80% | 70% | 90% | 80% | 70% | 90% | 80% | 70% |
| ILFB iter-1 | 91.90% | 76.84% | 77.77% | 100.0% | 99.99% | 98.67% | 97.12% | 81.34% | 75.57% |
| ILFB iter-2 | 96.05% | 91.95% | 79.12% | 100.0% | 99.85% | 99.10% | 97.33% | 88.11% | 76.45% |
| ILFB iter-3 | 99.15% | 96.14% | 85.66% | 100.0% | 99.91% | 99.69% | 97.43% | 89.48% | 86.63% |
| ILFB iter-4 | 100.0% | 99.67% | 91.51% | 100.0% | 99.96% | 99.75% | 97.53% | 92.89% | 82.15% |
| ILFB iter-5 | 100.0% | 100.0% | 97.00% | 100.0% | 99.99% | 99.95% | 98.14% | 92.94% | 94.02% |

periment runs for $80$ epochs. For each iteration of update using our ILFB algorithm, since less samples are used at each epoch, we adjust the total epoch number accordingly, so that the training for every iteration uses the same number of SGD updates. We set $\tau$ to be $5\%$ less than the true ratio of "good" samples.

**Results**  Table 1 shows the results for the baseline, oracle and our methods. Please see the caption thereof. We observe significant improvement over the baseline under all experiments. In Appendix B, we provide a comparison of using different initialization methods, where ILFB performs better than its counterpart with random initialization. In Appendix E, we provide more experimental results illustrating that ILFB is not sensitive to mis-specified $\tau$, the improvement of this iterative procedure compared with a single iteration, the advantage over other outlier removal options, and its capability of handling random label noise.

## 7   ILFB FOR DEEP GENERATIVE MODELS WITH MIXED TRAINING DATA

**Experimental setting**  We consider training a GAN – specifically, the DC-GAN architecture Radford et al. (2015) – to generate images similar to those from a *good* dataset, but when the training data given to it contains some fraction of the samples from a different *bad* dataset. All images are unlabeled, and we do not know which training sample comes from which dataset. We investigated the efficacy of our approach in three such settings:
*(a)* When the good dataset is the Celeb-A face images, and the bad dataset is CIFAR-10.
*(b)* When the good dataset is the CIFAR-10 face images, and the bad dataset is Celeb-A.
*(c)* When the good dataset is MNIST digits, and the bad is Fashion-MNIST Xiao et al. (2017).
For each of these, we consider different fractions of bad samples in the training data, evaluate their effect on standard GAN training, and then the efficacy of our approach as we execute it for upto 5 iterations.

**Algorithm**  Recall that training a GAN consists of updating the weights of both a generator network and a discriminator network; our model $\theta$ is the parameters of *both* of these networks. Unlike the classification seeing, we now need to develop a loss $f_\theta(s)$ – and unlike in the simple Gaussian mixture model setting we do not have explicit access to a likelihood function for the generative model. Our crucial innovation is to use the loss[7] at the output of the discriminator – the same one used to train the discriminator – as the $f_\theta(s)$. The algorithm starts by training on all samples, and then alternates between picking training samples with the smallest discriminator loss, and retraining

---

[7]Notice that for different GAN architectures, the loss function for training the discriminator varies, however, we can always find a surrogate loss by modifying the original loss function, and use the loss of the discriminator for real images as the surrogate loss for ILFB .

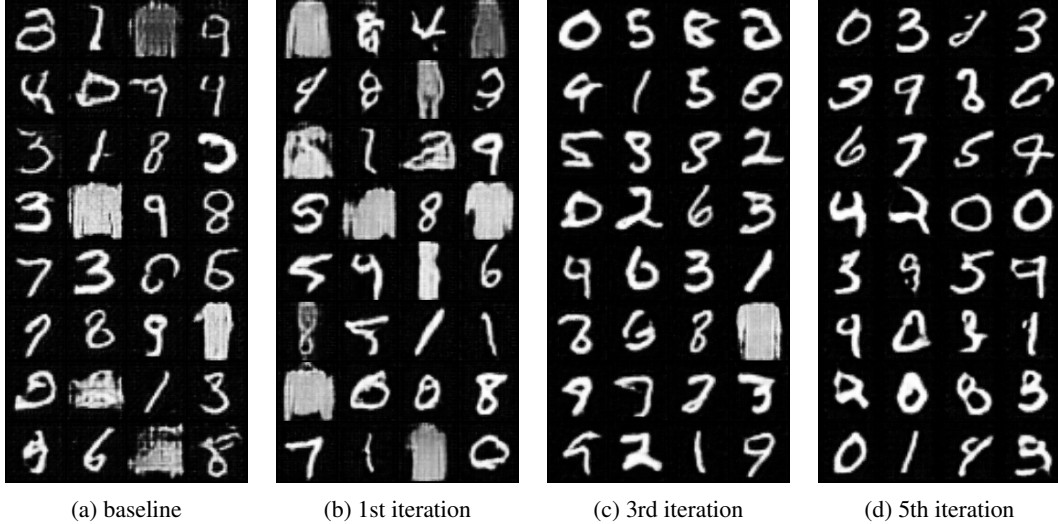

(a) baseline        (b) 1st iteration        (c) 3rd iteration        (d) 5th iteration

Figure 1: **Qualitative performance of ILFB for GANs:** We apply ILFB to a dataset of 80% MNIST "good" images + 20% Fashion-MNIST "bad". The panels show the fake images from 32 randomly chosen (and then fixed) latent vectors, as ILFB iterations update the GAN weights. Baseline is the standard training of fitting to all samples. We can see that the baseline generates both digit and fashion images, but by the 5th iteration it hones in on digit images.

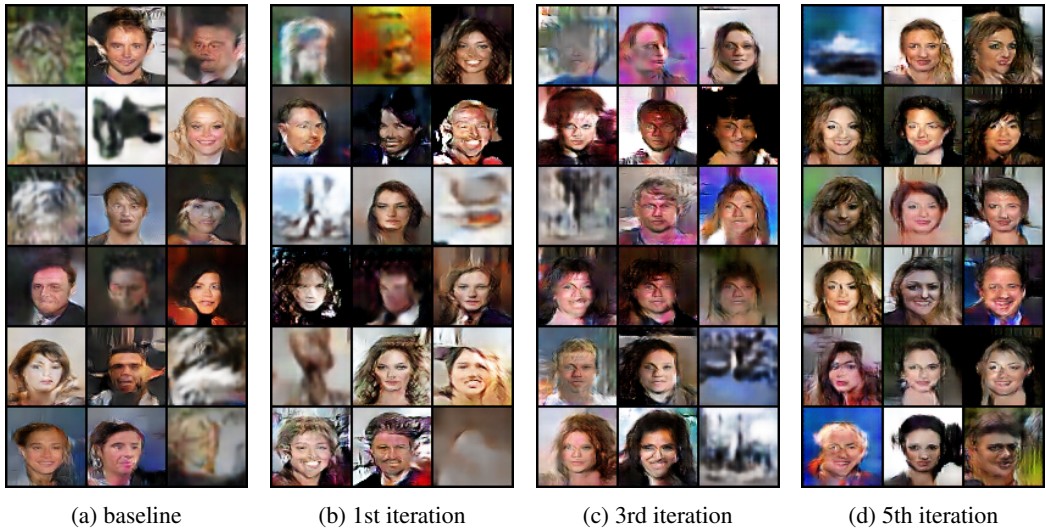

(a) baseline        (b) 1st iteration        (c) 3rd iteration        (d) 5th iteration

Figure 2: **Qualitative performance of ILFB for GANs:** Given training data consisting of 70% CelebA "good" images + 30% CIFAR-10 "bad" images, the four panels above each show the performance after iterations of our ILFB algorithm. First we choose 18 random vectors in latent pace, and fix them for all iterations. In each iteration, we retrain the GAN on corresponding selected samples to get new weighs, which are then used to generate the 18 fake images. Baseline refers to the standard training where we fit to all the samples (this is also our initialization). Visually, we can see that the generator is able to improve its generation quality and only generate face-like images after the 5th iteration.

the model on these picked samples. Here training means updating both the generator and discriminator network weights, which is done via SGD. Again, we set $\tau$ to be $5\%$ less than the true ratio of "good" samples.

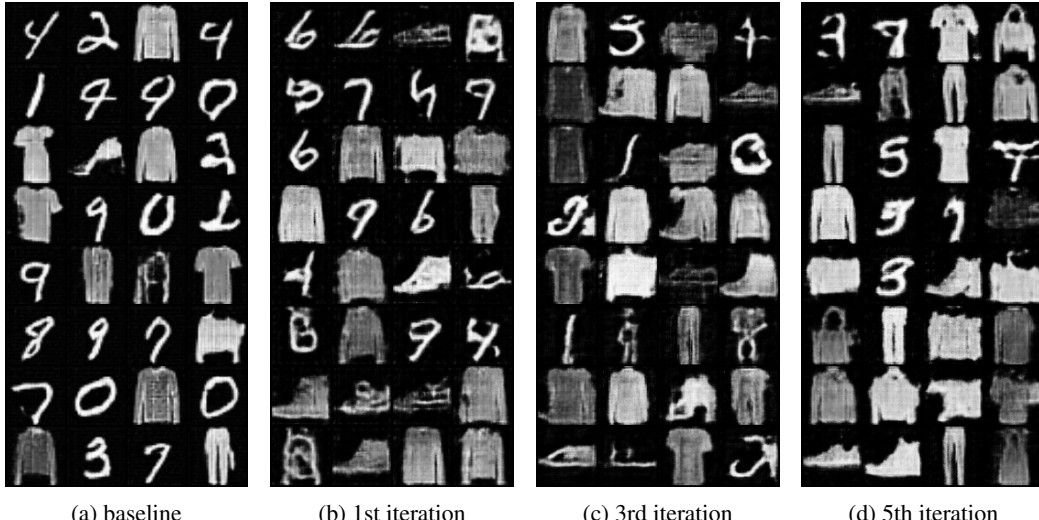

| (a) baseline | (b) 1st iteration | (c) 3rd iteration | (d) 5th iteration |

Figure 3: **Illustrative failure case:** This figure shows that when the fraction of "bad" samples is too large, ILFB cannot clean them out. The setting is exactly the same as in Figure 1, but now with 60% MNIST "good" images + 40% Fashion-MNIST "bad" images. We can see that now the $5^{th}$ iteration still retains the fake fashion images. Please see the discussion section for some intuition on this.

**Results** We evaluate the effect of bad training samples, and the improvement by our approach, in two ways: quantitatively in Table 2 and qualitatively in Figures 1 and 2. Figure 3 illustrates a failure case, when the bad set is too big, which we feel gives insight into what is going on. Please see the respective captions. The failure example happens for MNIST-Fashion dataset when ratio of MNIST images is $60\%$. In fact, for every iteration, ILFB selects all images from FashionMNIST (which counts for $0.4n$) and $0.15n$ MNIST images (the least number of MNIST images the algorithm can select). See Fig. 3 for qualitative evaluation of the results. In Appendix E, we provide more experimental results illustrating that ILFB is not sensitive to mis-specified $\tau$ and the advantage over other outlier removal options.

## 8 DISCUSSION

We demonstrated, both theoretically for simpler settings, and empirically for more challenging neural network ones, the merit of iteratively fitting to the best samples – in both supervised and unsupervised problems. The underlying **intuition** is that ILFB can focus in on the "good" samples, provided it is properly initailized; and when the number of bad samples is significant but not overwhelming, initial fitting on all the data is good enough to get started. We now add several discussions on the merits and otherwise of our results:

*(1)* One way to view the effect of ILFB in the GAN setting: a discriminator is likely least certain on the smallest modes of a distribution, and our process of dropping training samples based on this results in the elimination of smaller modes, which correspond to modes from the bad samples. This is illustrated by the failure case in Fig. 3, where the presence of a high fraction of "bad" fashion-MNIST samples makes their modes comparable to those of the "good" MNIST.

*(2)* Retaining samples best fit by the current model is inherently local, in the sense that samples with large errors are ignored. Finding a good initialization to start with is thus important; without this we may have bad performance. When the size of the good samples is comparatively large, an initial fit on all the samples is good enough; otherwise, we may use other alternatives to initialize the model, e.g., fit the model on a smaller dataset with clean data.

*(3)* The ILFB algorithm is simple and efficient enough to be applied to most modern machine learning tasks. When there exists extra computation power or knowledge on the specific dataset, it is certainly possible to improve based on ILFB . However, we believe ILFB is simple and general enough, and may serve as a strong result when more complicated outlier detection or noisy label algorithms are proposed.

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

# A EFFECT OF MIS-SPECIFICATION

## A.1 MIS-SPECIFICATION FOR A CLEAN DATASET

We test the performance of ILFB under mis-specified $\tau$ when the dataset is clean, i.e., there is only a single component. We choose random $\theta^\star \in \mathbb{R}^d$ with unit norm, the mis-specified $\tau$ is set as $0.95$, dimension $d = 100$, and sample size $n \in \{500, 1000, 1500\}$, $T = 30$. As shown in Fig. 4, as $\sigma$ gets larger, the gap between ILFB and OLS estimator (in terms of $\ell_2$ distance) increases linearly.

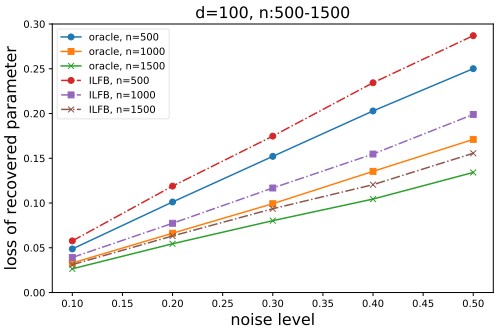

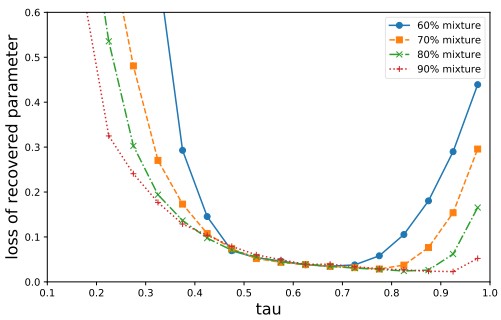

Figure 4: Effect of mis-specification for linear regression model

Figure 5: Effect of mis-specification for mixed linear regression

We further show the performance of ILFB as sample size increases. In Fig. 6, the asymptotic performance under different noise levels is shown. For the clean dataset, ILFB is observed to give consistent estimate. We observe similar performance in the Gaussian setting as well.

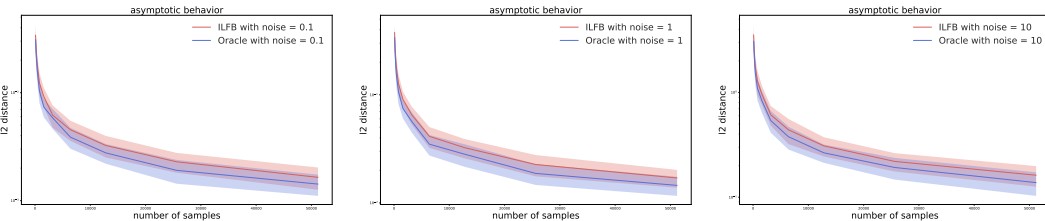

Figure 6: Consistency of ILFB linear regression under different noise levels ($\sigma = 0.1, 1, 10$ from left to right).

## A.2 MIS-SPECIFICATION FOR A NOISY DATASET

In this part, we test the performance when using different $\tau$ values for a noisy dataset. Again, we present the result for the mixed linear regression setting. We run ILFB for $10$ iterations and the results are based on an average of $50$ random runs. In Fig. 5, we see that the performance is not sensitive to the mis-specified $\tau$ in a large region. Interestingly, ILFB still performs well when $\tau$ is slightly larger than the true ratio of the component we are interested in.

# B RANDOM INITIALIZATION FOR CLASSIFICATION

We study the effect of initialization for the classification problem using MNIST (5%) dataset. In Table 3, we present the accuracy after 5-th iteration of training for both random initialization and the original ILFB methods. Even with random initialization, the performance is consistently better than the baseline, however, the improvement of accuracy is smaller when compared to the original ILFB , which uses all samples for training as initialization.

| $\dfrac{\text{\# good}}{\text{\# total}}$ | 60% | 70% | 80% | 90% |
|---|---|---|---|---|
| **Baseline** | 70.26 | 85.95 | 92.62 | 93.88 |
| **Random Initialization + ILFB** | 82.05 | 91.48 | 93.80 | 93.79 |
| **ILFB** | 90.00 | 92.09 | 94.30 | 94.41 |

Table 3: MNIST-5% dataset with systematic label error using ILFB with different initialization methods. The accuracy at the 5-th iteration is shown for both initialization methods. Results are averaged over 5 runs.

## C  LINEAR REGRESSION MODEL

Since we focus on recovering the parameters of one mixture component from the mixed linear regression model, we refer to the targeted component as the 'good' component, and the rest are 'bad' components. More specifically, the full set of samples $S = [n]$ can be splitted into a set of "good" samples $S_{\text{good}}$, and $m$ sets of "bad" samples $\cup_{j \in [m]} S_{\text{bad}}^j$. The response variables $y_i$ are given by:

$$y_i = \begin{cases} x_i^\top \theta^\star + \epsilon_i, & \text{if } i \in S_{\text{good}} \\ x_i^\top \theta^{(j)} + \epsilon_i, & \text{if } i \in S_{\text{bad}}^j \end{cases}$$

where $\epsilon_i \sim \mathcal{N}(0, \sigma^2)$ is the additive noise, and $\theta^\star$ as well as all the $\theta^{(j)}$ are assumed to be unit norm vectors.

### C.1  PROOF FOR UPDATE STEPS IN LINEAR REGRESSION MODEL

For simplicity of the notation, we present the proof by assuming $\mathbf{y}_i$ for bad samples have the form $\mathbf{y}_i = \mathbf{n}_i + \mathbf{e}_i$, where $\mathbf{n}_i$ corresponds to $\mathbf{x}_i^\top \theta^{(j)}$, such that $i \in S_{\text{bad}}^j$.

Let $\theta_t$ be the current learned parameter. $\theta_{t+1}$ is learned by using one iteration in Algorithm 1. Namely, we first select a subset $S_t$ of size $\tau n$ with the smallest loss $(\mathbf{y}_i - \mathbf{x}_i^\top \theta_t)^2$, then re-calculate the MLE estimator only using those samples. Denote $\mathbf{W}_t$ as the diagonal matrix whose diagonal entry $\mathbf{W}_{ii}$ equals 1 when the $i$-th sample is in set $S_t$, otherwise equals 0. Then,

$$\theta_{t+1} = \left(\mathbf{X}^\top \mathbf{W}_t \mathbf{X}\right)^{-1} \mathbf{X}^\top \mathbf{W}_t \mathbf{y}, \tag{3}$$

where we used the fact that $\mathbf{W}_t^2 = \mathbf{W}_t$. Notice that $\mathbf{W}^\star$ is the ground truth for $\mathbf{W}_t$ (accordingly, define $S^\star$ for the ground truth of $S_t$). For clearness of the presentation, we ignore the subscript $t$ when there is no ambiguation. Then, according to the definition of $\mathbf{y}$ and reorganizing the terms, (3) can be expanded as

$$\begin{aligned}
\theta_{t+1} - \theta^\star &= \left(\mathbf{X}^\top \mathbf{W} \mathbf{X}\right)^{-1} \mathbf{X}^\top \mathbf{W} \left(\mathbf{W}^\star \mathbf{X} \theta^\star + (\mathbf{I} - \mathbf{W}^\star) \mathbf{n} + \mathbf{e}\right) - \theta^\star \\
&= \left(\mathbf{X}^\top \mathbf{W} \mathbf{X}\right)^{-1} \left(\mathbf{X}^\top \mathbf{W} \mathbf{W}^\star \mathbf{X} \theta^\star + \mathbf{X}^\top \mathbf{W} \mathbf{n} - \mathbf{X}^\top \mathbf{W} \mathbf{W}^\star \mathbf{n} - \mathbf{X}^\top \mathbf{W} \mathbf{X} \theta^\star + \mathbf{X}^\top \mathbf{W} \mathbf{e}\right) \\
&= \left(\mathbf{X}^\top \mathbf{W} \mathbf{X}\right)^{-1} \mathbf{X}^\top \left(\mathbf{W} \mathbf{W}^\star - \mathbf{W}\right) \left(\mathbf{X} \theta^\star - \mathbf{n} - \mathbf{e}\right) + \left(\mathbf{X}^\top \mathbf{W} \mathbf{X}\right)^{-1} \mathbf{X}^\top \mathbf{W} \mathbf{W}^\star \mathbf{e}
\end{aligned}$$

where as we addressed at the beginning of this section, for vector $\mathbf{n}$, given $i \in S_{\text{bad}}$, $\mathbf{n}_i = \mathbf{x}_i^\top \theta^{(j)}$ for some $j \in [m]$. Therefore,

$$\begin{aligned}
&\|\theta_{t+1} - \theta^\star\|_2 \\
&= \left(\mathbf{X}^\top \mathbf{W} \mathbf{X}\right)^{-1} \mathbf{X}^\top \left(\mathbf{W} \mathbf{W}^\star - \mathbf{W}\right) \left(\mathbf{X} \theta^\star - \mathbf{X} \theta_t + \mathbf{X} \theta_t - \mathbf{n} - \mathbf{e}\right) + \left(\mathbf{X}^\top \mathbf{W} \mathbf{X}\right)^{-1} \mathbf{X}^\top \mathbf{W} \mathbf{W}^\star \mathbf{e} \\
&\leq \underbrace{\left\|\left(\mathbf{X}^\top \mathbf{W} \mathbf{X}\right)^{-1}\right\|_2}_{T_1} \cdot \underbrace{\left\|\mathbf{X}^\top \left(\mathbf{W} \mathbf{W}^\star - \mathbf{W}\right) \left(\mathbf{X} \theta^\star - \mathbf{n} - \mathbf{e}\right)\right\|_2}_{T_2} + \underbrace{\left\|\left(\mathbf{X}^\top \mathbf{W} \mathbf{X}\right)^{-1} \mathbf{X}^\top \mathbf{W} \mathbf{W}^\star \mathbf{e}\right\|_2}_{T_3}
\end{aligned} \tag{4}$$

We bound the three terms on the right hand side of (4) separately.

**Helpful lemmas**

**Lemma 3** (Sub-Gamma property for the $q$-quantile). *Let $n \geq 3$, let $X_{(1)} \geq \cdots \geq X_{(n)}$ be the order statistics of the absolute value of standard Gaussian samples. $qn$ is an integer for some constant $q \in (0, 1)$. Let $v_n = \frac{8}{qn \log 2}$. For all $0 \leq \lambda < \frac{qn}{2\sqrt{v_n}}$,*

$$\log \mathcal{E} \left[ e^{\lambda \left( X_{(qn)} - \mathcal{E}\left[ X_{(qn)} \right] \right)} \right] \leq \frac{v_n \lambda^2}{2 \left( 1 - 2\lambda \sqrt{\frac{v_n}{qn}} \right)} \tag{5}$$

*This shows the sub-gamma property of $X_{(qn)}$, and as a result, for all $t > 0$, and constant $\frac{1}{2} < q < 1$,*

$$\mathcal{E}\left[ X_{(qn)} \right] = \psi^{-1} \left( \frac{1-q}{2} \right) \tag{6}$$

$$\mathcal{P}\left[ \left| X_{(qn)} - \mathcal{E}\left[ X_{(qn)} \right] \right| \geq \sqrt{2c_1 v_n \log n} + 2c_1 \sqrt{\frac{v_n}{qn}} \log n \right] \leq n^{-c_1}. \tag{7}$$

*This result is generalized from the result in Boucheron et al. (2012).*

**Lemma 4.** *Suppose we have two Gaussian distributions $\mathcal{D}_1 = \mathcal{N}(0, \Delta^2), \mathcal{D}_2 = \mathcal{N}(0, 1)$. We have $fn$ i.i.d. samples from $\mathcal{D}_1$ and $(1-f)n$ i.i.d. samples from $\mathcal{D}_2$. Denote the set of the top $\bar{f}n$ samples with smallest abstract values as $S_{\bar{f}n}$, where $\bar{f} \leq f$. Then, with high probability, for $\Delta \leq 1$, at most $\left( c \max \left\{ \Delta (1-f) n \sqrt{\log n}, \log n \right\} \right)$ samples in $S_{\bar{f}n}$ are from $\mathcal{D}_2$.*

*Proof of Lemma 4.* First, let us consider the probability of a random variable from $\mathcal{D}_2$ has smaller abstract value than a random variable from $\mathcal{D}_1$. This would give us a rough idea about the occupancy of the samples from $\mathcal{D}_2$ in the top set. Denote the two variables from $\mathcal{D}_1, \mathcal{D}_2$ as $u_1, u_2$ respectively. Then, with basic calculations, we have

$$\mathcal{P}\left[ |u_2| \leq |u_1| \right]$$

$$= \int_0^\infty \mathcal{P}\left[ |u_1| \geq s \right] \frac{2}{\sqrt{2\pi}} e^{-\frac{s^2}{2}} ds$$

$$\leq \int_0^\infty 2e^{-\frac{s^2}{2\Delta^2}} \frac{2}{\sqrt{2\pi}} e^{-\frac{s^2}{2}} ds$$

$$= 2\frac{\Delta}{\sqrt{1+\Delta^2}}.$$

This calculation implies that if given same number of samples from $\mathcal{D}_1$ and $\mathcal{D}_2$, it is reasonable to expect that only around $2\Delta$ fraction of the top set samples are from $\mathcal{D}_2$, when $\Delta$ is small. In the original setting where we have $\bar{f}n$ samples from $\mathcal{D}_1$ and $(1 - \bar{f}) n$ from $\mathcal{D}_2$, we should expect around $2\Delta \frac{1-\bar{f}}{\bar{f}} \bar{f}n = 2\Delta (1 - \bar{f})$ samples in the top set are from $\mathcal{D}_2$.

Now, we aim for a rigorous proof of the result with good concentration. Our idea is to first show concentration result for the maximum value from distribution $\mathcal{D}_1$, and then, we show the high probability upper bound for the number of samples from $\mathcal{D}_2$ that are less than the maximum value.

**Step I – the sample with maximum abstract value from $\mathcal{D}_1$.** We know that for random normal i.i.d. Gaussian variables $x_i, i \in [n]$,

$$\mathcal{P}\left[ \max_{i \in [n]} |x_i| \geq \sqrt{2 \log 2n} + t \right] \leq 2e^{-\frac{t^2}{2}}.$$

Therefore, for $fn$ samples from $\mathcal{D}_1$, with high probability, the maximum abstract value is in the order of $O(\sqrt{\log n}\Delta)$.

**Step II.** On the other hand, for a random $u_2 \sim \mathcal{D}_2$, we know that for small positive values $\delta = c\sqrt{\log n}\Delta$, $\mathcal{P}\left[ |u_2| \leq \delta \right] \leq \sqrt{\frac{2}{\pi}}\delta$ gives a tight upper bound. Let $\mathcal{M}_{\delta,i}$ be the event *sample $u_i$ from*

$\mathcal{D}_2$ *has abstract value less than* $\delta$, and a Bernoulli random variable $m_{i,\delta}$ that is the indicator of event $\mathcal{M}_{\delta,i}$ holds or not. Then,

$$\mathcal{E}\left[\sum_{i=1}^{(1-f)n} m_{i,\delta}\right] \leq \sqrt{\frac{2}{\pi}}\delta\left(1-f\right)n.$$

For independent random variable $x_i$s, $i \in [\tilde{n}]$ that lie in interval $[0,1]$, with $X = \sum_i x_i$ and $\mu = \mathcal{E}[X]$, Chernoff's inequality tells us

$$\mathcal{P}\left[X \geq (1+\gamma)\,\mu\right] \leq e^{-\frac{\mu\gamma^2}{3}}, \quad \forall \gamma \in [0,1]$$
$$\mathcal{P}\left[X \geq (1+\gamma)\,\mu\right] \leq e^{-\frac{\mu\gamma}{3}}, \quad \forall \gamma > 1$$

As a consequence, for $m_{i,\delta}$s, we have

$$\mathcal{P}\left[\sum_{i=1}^{(1-f)n} m_{i,\delta} \geq (1+\gamma)\sqrt{\frac{2}{\pi}}\delta\left(1-f\right)n\right] \leq e^{-\frac{\gamma^2\sqrt{\frac{2}{\pi}}\delta(1-f)n}{3}}, \quad \forall \gamma \in [0,1]$$

$$\mathcal{P}\left[\sum_{i=1}^{(1-f)n} m_{i,\delta} \geq (1+\gamma)\sqrt{\frac{2}{\pi}}\delta\left(1-f\right)n\right] \leq e^{-\frac{\gamma\sqrt{\frac{2}{\pi}}\delta(1-f)n}{3}}, \quad \forall \gamma > 1.$$

For the first case, where $\gamma \in [0,1]$, we can set $\gamma = c\sqrt{\frac{\log n}{\delta(1-f)n}}$ to get high probability guarantee. The constraint on $\gamma$ requires $\delta n > c\log n$ for some fixed $c$. On the contrary, when this is violated, i.e., when $\delta$ is much smaller, then, by the Chernoff bound for the case $\gamma > 1$, we can set $\gamma = \frac{c\log n}{\delta(1-f)n}$.

**Combining Step I and Step II.**   To summarize, for some fixed constant $c$, with high probability:

- For $\Delta > \frac{c\sqrt{\log n}}{n}$, at most $2c\delta\left(1-f\right)n = c\sqrt{\log n}\Delta\left(1-f\right)n$ samples in $S_{\bar{f}n}$ are from $\mathcal{D}_2$.

- For $\Delta \leq \frac{c\sqrt{\log n}}{n}$, at most $(1+\gamma)\,\delta\left(1-f\right)n = c\log n$ samples in $S_{\bar{f}n}$ are from $\mathcal{D}_2$.

■

**Lemma 5.** *Assume* $\tau < \tau^\star$ *(* $\tau/\tau^\star = c_\tau < 1$ *is a constant), with high probability, we have:*

$$\sigma_{\min}\left(\mathbf{X}^\top \mathbf{W}\mathbf{X}\right) \geq c_0|S_t \cap S^\star| + \sum_{j \in [m]} c_j|S_t \cap S^j| \geq c\tau n, \tag{8}$$

*where* $c_0, \cdots, c_m, c$ *are constants.*

*Proof of Lemma 5.* We focus on finding concentration results for the smallest eigenvalue of $\mathbf{X}^\top \mathbf{W}\mathbf{X}$. Rewrite $\mathbf{X}^\top \mathbf{W}\mathbf{X}$ as

$$\mathbf{X}^\top \mathbf{W}\mathbf{X} = \sum_i \mathbf{x}_i \mathbf{x}_i^\top \mathbf{1}\left[l_{i,t} \in \arg\tau n\text{-}\min\left(l_{1,t}, \cdots, l_{n,t}\right)\right]$$

$$= \sum_i \mathbf{x}_i \mathbf{x}_i^\top \mathbf{1}\left[l_{i,t} \in \arg\tau n\text{-}\min\left(l_{1,t}, \cdots, l_{n,t}\right)\right] \mathbf{1}\left[i \in S^\star\right]$$

$$+ \sum_{j \in [m]} \sum_i \mathbf{x}_i \mathbf{x}_i^\top \mathbf{1}\left[l_{i,t} \in \arg\tau n\text{-}\min\left(l_{1,t}, \cdots, l_{n,t}\right)\right] \mathbf{1}\left[i \in S^j\right] \tag{9}$$

where $\mathbf{1}[\cdot]$ is the indicator function. Note that the values $\{l_{i,t}\}_{i=1}^n$ are defined as:

$$l_{i,t} = \left(\mathbf{y}_i - \mathbf{x}_i^\top \theta_t\right)^2.$$

More specifically, consider the model setting, we have:

$$l_{i,t} = \left[\mathbf{x}_i^\top (\theta^\star - \theta_t) + \epsilon_i\right]^2, \quad i \in S^\star$$
$$l_{i,t} = \left(\mathbf{x}_i^\top (\theta^j - \theta_t) + \epsilon_i\right)^2, \quad i \in S^j, j \in [m]$$

For all $i \in S^{\star}$, $l_{i,t}$s have the same distribution as $\delta\mu$, where $\mu$ is a $\chi^2(1)$ random variable, and $\delta$ is the magnitude $\delta = \|\theta^{\star} - \theta_t\|_2^2 + \sigma^2$. The bad samples $i \in [n]\backslash S^{\star}$, for $i \in S_{\text{bad}}^j$ have the same distribution as $(\sigma^2 + \|\theta^{(j)} - \theta_t\|_2^2)\mu$. Also, the assumption in $\theta_t$ assumes that the magnitude of the good distribution is smaller. Applying the results in Lemma 3 and Lemma 4, assuming the condition in Theorem 1 holds, for given $\theta_t, \theta^{\star}$, there exists a constant $c_1$ such that the separation value for deciding top-$\tau n$ or not is lower bounded by $c\delta$ with high probability. For a fixed $c_0$, we consider the following quantity:

$$\mathbf{\Sigma}(c_0) = \mathcal{E}\left[\mathbf{x}_i\mathbf{x}_i^{\top} \mid |\mathbf{x}_i^{\top}(\theta^{\star} - \theta_t) + \epsilon_i| \leq c_0\sqrt{\|\theta^{\star} - \theta_t\|_2^2 + \sigma^2}\right].$$

By the geometry of the constraint, we have that

$$\sigma_{\min}(\mathbf{\Sigma}(c)) \geq \int_{-c}^{c} s^2 \frac{1}{\sqrt{2\pi}} e^{-\frac{s^2}{2}} ds = 1 - 2\psi(c) - ce^{-\frac{c^2}{2}}$$

In other words, the lower bound we have for the smallest eigenvalue of the expectation of (9) monotonically increases with $c$, and can be substituted with another constant $2c_0$ (notice that previously, we have a lower bound on the separation value). Therefore, considering the sub-exponential property, we get the concentration result based on the above expected value:

$$\sigma_{\min}\left(\sum_i \mathbf{x}_i\mathbf{x}_i^{\top}\mathbf{1}\left[l_{i,t} \in \arg \tau n\text{-}\min(l_{1,t}, \cdots, l_{n,t})\right]\mathbf{1}\left[i \in S^{\star}\right]\right) \geq \frac{c_0}{2}|S_t \cap S^{\star}|,$$

Similarly, since $\|\theta^j - \theta_t\| \leq 3$, we have a lower bound $O(|S_t \cap S^j|)$ for $i \in S_t \cap S^j$ as well, as long as the set $|S_t \cap S^j| \geq c\log^2 n$. In any case, we have the minimum eigenvalue is lower bounded by $c\tau n$.

∎

$T_2$ **in (4)**     $T_2$ can be bounded as follows:

$$
\begin{aligned}
T_2^2 &= \left\|\sum_{i \in S_t\backslash S^{\star}}\left(\mathbf{x}_i^{\top}\theta^{\star} - \mathbf{n}_i - \mathbf{e}_i\right)\mathbf{x}_i\right\|_2^2 \\
&= (\mathbf{X}\theta^{\star} - \mathbf{n} - \mathbf{e})^{\top}(\mathbf{W} - \mathbf{W}\mathbf{W}^{\star})\mathbf{X}\mathbf{X}^{\top}(\mathbf{W} - \mathbf{W}\mathbf{W}^{\star})(\mathbf{X}\theta^{\star} - \mathbf{n} - \mathbf{e}) \\
&\leq \sigma\left((\mathbf{W} - \mathbf{W}\mathbf{W}^{\star})\mathbf{X}\mathbf{X}^{\top}(\mathbf{W} - \mathbf{W}\mathbf{W}^{\star})\right)(\mathbf{X}\theta^{\star} - \mathbf{n} - \mathbf{e})^{\top}(\mathbf{W} - \mathbf{W}\mathbf{W}^{\star})(\mathbf{X}\theta^{\star} - \mathbf{n} - \mathbf{e}) \\
&= \sigma\left((\mathbf{W} - \mathbf{W}\mathbf{W}^{\star})\mathbf{X}\mathbf{X}^{\top}(\mathbf{W} - \mathbf{W}\mathbf{W}^{\star})\right)\sum_{j \in [m]}\sum_{k \in S_t \cap S^{(j)}\backslash S^{\star}}\left(\mathbf{x}_k^{\top}\theta^{\star} - \mathbf{x}_k^{\top}\theta^{(j)} - \mathbf{e}_k\right)^2 \\
&\leq \sigma\left((\mathbf{W} - \mathbf{W}\mathbf{W}^{\star})\mathbf{X}\mathbf{X}^{\top}(\mathbf{W} - \mathbf{W}\mathbf{W}^{\star})\right)\sum_{j \in [m]}\sum_{k \in S_t \cap S^{(j)}\backslash S^{\star}}\left[\left(\mathbf{x}_k^{\top}(\theta^{\star} - \theta_t)\right)^2 + \left(\mathbf{x}_k^{\top}(\theta_t - \theta^{(j)}) - \mathbf{e}_k\right)^2\right] \\
&\leq \sigma\left((\mathbf{W} - \mathbf{W}\mathbf{W}^{\star})\mathbf{X}\mathbf{X}^{\top}(\mathbf{W} - \mathbf{W}\mathbf{W}^{\star})\right)^2\|\theta_t - \theta^{\star}\|_2^2 \\
&\quad + c\sigma\left((\mathbf{W} - \mathbf{W}\mathbf{W}^{\star})\mathbf{X}\mathbf{X}^{\top}(\mathbf{W} - \mathbf{W}\mathbf{W}^{\star})\right)|S_t\backslash S^{\star}|\left(\|\theta_t - \theta^{\star}\|_2^2 + \sigma^2\right) \\
&\leq c|S_t\backslash S^{\star}|^2\left(\|\theta_t - \theta^{\star}\|_2^2 + \sigma^2\right)
\end{aligned}
$$

for $|S_t\backslash S^{\star}| \geq c\log^2 n$ (when $|S_t\backslash S^{\star}| < c\log^2 n$, the final convergence result still holds true). As a consequence,

$$T_2 \leq c|S_t\backslash S^{\star}|(\|\theta_t - \theta^{\star}\|_2 + \sigma). \tag{10}$$

$T_3$ **in (4)**     For this part, we can reuse the result we proved for $T_1$. Also, notice that with high probability:

$$\left\|\mathbf{X}^{\top}\mathbf{W}\mathbf{W}^{\star}\mathbf{e}\right\|_2^2 = \mathbf{e}^{\top}\mathbf{W}^{\star}\mathbf{W}\mathbf{X}\mathbf{X}^{\top}\mathbf{W}\mathbf{W}^{\star}\mathbf{e} \leq cd\log nn\sigma^2 \tag{11}$$

**Putting things together**  Combining Lemma 5, (10), and (11), we have:

$$\|\theta_{t+1} - \theta^\star\|_2$$
$$\leq \frac{c'|S_t\backslash S^\star|(\|\theta_t - \theta^\star\|_2 + \sigma)}{\tau n} + \frac{c\sqrt{nd\log n}}{\tau n}\sigma$$
$$\leq c_3\|\theta_t - \theta^\star\|_2 + (c_3 + c_4)\sigma.$$

We require $n \geq \frac{cd\log d}{c_4^2\tau}$. Also, according to Lemma 4, given $\|\theta_t - \theta^\star\|_2 \leq c_3 c\frac{\tau}{1-\tau}\min_{j\in[m]}\|\theta_t - \theta^j\|_2 - \sqrt{1 - \left(\frac{c_3 c\tau}{1-\tau}\right)^2}\sigma$ (for $c_3 \in (0, 1)$), then with high probability, $|S_t\backslash S^\star| \leq \frac{c_3}{c'}\tau n$. Notice that for small $\tau$, the noise should not be too large. Otherwise, even if $\theta_t$ is very close to $\theta^\star$, because of the noise and the high density of bad samples, $|S_t\backslash S^\star|$ would still be quite large, and the update will not converge.

## C.2  SIMULATIONS

### C.2.1  ILFB ON MIXED LINEAR REGRESSION

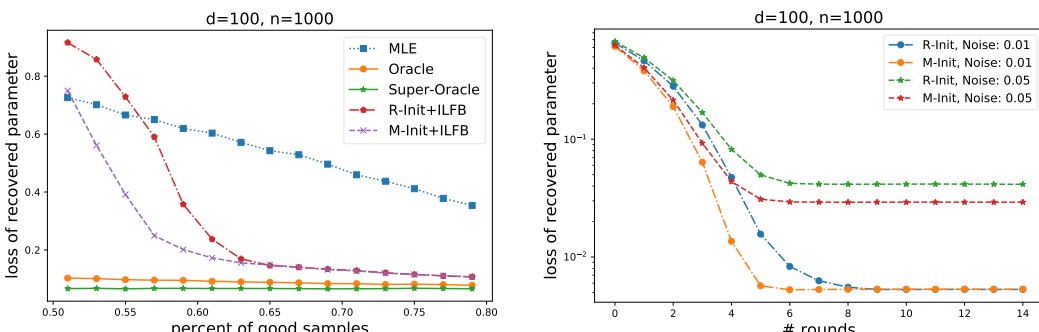

Figure 7: **Left:** $\ell_2$ loss of recovered parameter to the true parameter for (a) MLE: naive MLE/OLS estimator; (b) Oracle: MLE estimator for the subset of good samples; (c) Super-Oracle: MLE estimator for the whole set of samples given the correct output for bad samples; (d) R-Init+ILFB: our algorithm with random initialization; (e)M-Init+ILFB: our algorithm with MLE as initialization. $\sigma = 0.2$, systematic error setting. **Right:** Speed of convergence under systematic error setting, number of good samples is set as $600$. M-Init: ILFB using MLE initialization; R-Init: ILFB using random initialization.

We verify the performance of ILFB on mixed linear regression model via synthetic experiments. When constructing the dataset, we consider a special case where all bad samples are generated from a single $\beta_j$. The parameters for both the good and bad components are generated randomly on a unit sphere. First, in Fig. 7 (left), we test the performance when $\tau^\star$ varies in the interval $[0.5, 0.8]$ with $\sigma = 0.2$. We compare:

(1) *MLE*: All samples are treated as good samples to find the best parameter;

(2) *Oracle*: learn from the $\tau^\star n$ good samples given oracle access;

(3) *Super-Oracle*: learn from $n$ samples given oracle access;

(4) *R-Init+ILFB* : ILFB with random initialization;

(5) *M-Init+ILFB* : ILFB using *MLE* as initialization.

The parameter recovery ($y$-axis) is measured by $\|\theta_{\mathrm{alg}} - \theta^\star\|_2$. We can see that ILFB with MLE as initialization performs better than the random initialization counterpart, which shows the effectiveness of taking good initailizations. Next, we show experimentally the speed of convergence in Fig. 7 (right) with multiple levels of measurement noise (in the small noise regime). The plots show linear convergence of each update step, which matches the result in Theorem 1.

### C.2.2 ASYMPTOTIC PERFORMANCE

Next, we check the performance of ILFB asymptotically. Our theory implies that the final recovery will get to a noise ball centered at the true parameter, but does not ensure the algorithm goes to the true parameter, even when sample size goes to infinity. In this section, we show experimentally that this noise ball guarantee is indeed tight, i.e., one should not expect ILFB to give exact recovery even in the infinite sample case.

Our experimental setting is as follows: we consider mixed linear regression with two components, where $60\%$ of the samples comes from the interested component, and all $\mathbf{x}_i$s follow isotropic normal Gaussian distribution. The parameters for the two components are set as unit norm, and orthogonal to each other, with $d = 10$. We run ILFB for 15 iterations with $\tau$ set as $5\%$ less than that of the true ratio. Number of samples varies from 100 to 51200. Standard deviation is calculated based on 100 runs for each experiment.

Fig. 8 provides the comparison between ILFB and oracle performance under different noise regime, i.e., $\sigma = 0.1, 1, 10$. We observe the performance of ILFB is close to the oracle in the small noise regime, while when the noise is comparable to the signal, there exists a gap between ILFB and the oracle, as the result of ILFB falls within the noise ball and stops decreasing, while the oracle will go to the exact solution asymptotically. Similar performance appears in the large noise regime, however, for $n \leq 51200$, the gap between the two seems not too far apart, which is due to the dominance of the noise.

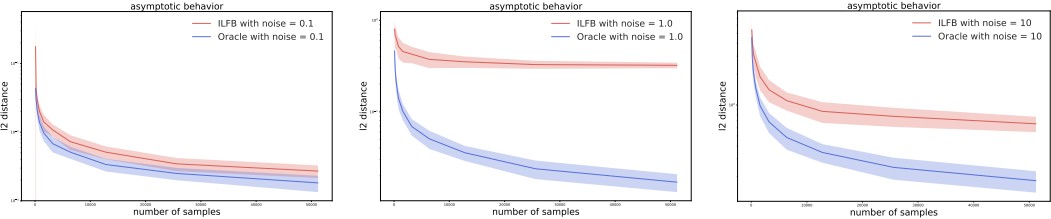

Figure 8: Performance of ILFB for mixed linear regression under different noise levels ($\sigma = 0.1, 1, 10$ from left to right). For small noise (left plot), the performance of ILFB decreases along with the oracle performance. When the noise gets larger (middle plot), we observe a gap between ILFB and the oracle when sample size $n$ increases. When the noise gets dominate the signal (right plot), the performance of ILFB and the oracle are not far apart for $n \leq 51200$, but the gap will get larger asymptotically. Notice that for all experiments, the standard deviation of the true signal is 1.

### C.3 INITIALIZATION FOR LINEAR REGRESSION MODEL

**Theorem 6 (initialization).** *Given the model described in (1), for some $j \in [m]$, define $d_m^j := \min_{l \in [m] \setminus \{j\}} \|\theta^{(j)} - \theta^{(l)}\|_2$, if $\sum_{l \in [m] \setminus \{j\}} \|\theta^{(j)} - \theta^{(l)}\|_2 |S^l| \leq \frac{n}{\kappa} d_m^j$ holds for some $\kappa$. We have:*

$$\|\theta^{(j)} - \theta_0\|_2 \leq \frac{1.1}{\kappa} d_m^j$$

*with high probability, where $\theta_0 = \theta_{\mathrm{OLS}}$, $n \geq c \max\{\frac{\kappa^2 d \sigma^2}{(d_m^j)^2}, \frac{m \kappa^2 \log(m\kappa)}{(d_m^j)^2}\}$.*

The assumption in Theorem 6 requires the total sum of distances of the parameters for samples from other components should not be too large. For a fixed $d_m^j$, if the distance for some $\theta^{(l)}$ becomes larger, it may become harder for the OLS estimator to be in the local region close to $\theta^{(j)}$. When the condition holds for some large $\kappa$, e.g., $\kappa > 2.2(1 + \frac{1-\tau}{c_1 c_3 \tau})$ and $\sigma < \frac{c_1 c_3 \tau d_m}{2\sqrt{(1-\tau)^2 - c_1^2 c_3^2 \tau^2}}$ when $C(\tau) < 1$ (or for some large $\tau$, $\kappa > 2.2$), following Theorem 1, the algorithm converges to a local $O(\sigma)$ ball around $\theta^{(j)}$.

*Proof.* We write out the OLS estimator as follows:

$$\begin{aligned}
\theta_{\mathrm{OLS}} &= \left(\mathbf{X}^\top\mathbf{X}\right)^{-1}\mathbf{X}^\top\mathbf{y} \\
&= \left(\mathbf{X}^\top\mathbf{X}\right)^{-1}\mathbf{X}^\top\left(\mathbf{W}^\star\mathbf{X}\theta^\star + (\mathbf{I}-\mathbf{W}^\star)\,\mathbf{n} + \mathbf{e}\right) \\
&= \left(\mathbf{X}^\top\mathbf{X}\right)^{-1}\mathbf{X}^\top\left(\mathbf{X}\theta^\star - (\mathbf{I}-\mathbf{W}^\star)\mathbf{X}\theta^\star + (\mathbf{I}-\mathbf{W}^\star)\mathbf{n} + \mathbf{e}\right) \\
&= \theta^\star - \left(\mathbf{X}^\top\mathbf{X}\right)^{-1}\mathbf{X}^\top\left(\mathbf{I}-\mathbf{W}^\star\right)\left(\mathbf{X}\theta^\star - \mathbf{n}\right) + \left(\mathbf{X}^\top\mathbf{X}\right)^{-1}\mathbf{X}^\top\mathbf{e}
\end{aligned}$$

Therefore, with high probability,

$$\|\theta_{\mathrm{OLS}} - \theta^\star\|_2$$

$$\leq \left\|\sum_{j=1}^{m}\sum_{i\in S_{\mathrm{bad}}^j}\left(\mathbf{X}^\top\mathbf{X}\right)^{-1}\mathbf{x}_i\mathbf{x}_i^\top\left(\theta^\star - \theta^{(j)}\right)\right\|_2 + \left\|\left(\mathbf{X}^\top\mathbf{X}\right)^{-1}\mathbf{X}^\top\mathbf{e}\right\|_2$$

$$\leq \frac{\sum_{j\in[m]}\|\theta^\star - \theta^{(j)}\|_2 \cdot |S_{\mathrm{bad}}^j| + c\sqrt{m(1-\tau)n\log n}}{n - c\sqrt{n\log n}} + \frac{\sqrt{(d + c\sqrt{d\log n})(n + c\sqrt{n\log n})}}{n - c\sqrt{n\log n}}\sigma$$

$$\leq \frac{\sum_{j\in[m]}\|\theta^\star - \theta^{(j)}\|_2 \cdot |S_{\mathrm{bad}}^j| + c\sqrt{m(1-\tau)n\log n}}{n}\left(1 + c\sqrt{\frac{\log n}{n}}\right) + \frac{\sqrt{2(d + c\sqrt{d\log n})}}{\sqrt{n}}\left(1 + c\sqrt{\frac{\log n}{n}}\right)\sigma$$

$$\leq \left(\frac{1}{\kappa}d_m + c\sqrt{\frac{m(1-\tau)\log n}{n}} + \frac{\sqrt{2(d + c\sqrt{d\log n})}}{\sqrt{n}}\sigma\right)\left(1 + c\sqrt{\frac{\log n}{n}}\right) \tag{12}$$

$$\leq \frac{1.1}{\kappa}d_m,$$

for $n \geq c\max\{\frac{\kappa^2 d\sigma^2}{d_m^2}, \frac{m\kappa^2\log(m\kappa)}{d_m^2}\}$, where in (12), we use the assumption $\sum_{j\in[m]}\|\theta^\star - \theta^{(j)}\|_2|S_{\mathrm{bad}}^j| \leq \frac{n}{\kappa}\min_{j\in[m]}\|\theta^\star - \theta^{(j)}\|_2$.

∎

# D    GAUSSIAN MIXTURE MODEL

Similar to Section C, we refer to the targeted component as the 'good' component and the rest are 'bad' components. More speifically, $S$ can be splitted into disjoint sets $S = S_{\mathrm{good}}\cup S_{\mathrm{bad}}^1\cup\cdots\cup S_{\mathrm{bad}}^m$, the samples follows:

$$\begin{aligned}
\mathbf{x}_i &\sim \mathcal{N}\left(\theta^\star, \sigma^2 I\right), &&\text{if } i \in S_{\mathrm{good}} \\
\mathbf{x}_i &\sim \mathcal{N}\left(\theta^{(j)}, \sigma^2 I\right), &&\text{if } i \in S_{\mathrm{bad}}^j
\end{aligned}$$

## D.1    PROOF FOR UPDATE STEPS IN GAUSSIAN MIXTURE MODEL

Similar to the proof for linear regression setting, we use $S_t$ to denote the set of selected samples according to parameter $\theta_t$. Then,

$$\theta_{t+1} = \frac{1}{\tau n}\sum_{i\in S_t}\mathbf{x}_i. \tag{13}$$

Assume $S^\star \subset S_{\mathrm{good}}$ to be the set of good samples that are closest to $\theta^\star$, and $|S^\star| = |S_t| = \tau n$. In other words, $S^\star$ is selected by growing a ball centered at $\theta^\star$, until $\tau n$ of the good samples fall into this ball, and we select them to be the set $S^\star$. Accordingly, let

$$\bar{\theta} = \frac{1}{\tau n}\sum_{i\in S^\star}\mathbf{x}_i. \tag{14}$$

With high probability, $\|\bar{\theta} - \theta^\star\|_2 \leq c\frac{\sqrt{d}\sigma\log n}{\sqrt{\tau n}} = o(\sigma\sqrt{d})$. For $i \in S_{\mathrm{bad}} \cap S_t$, the norm of $\mathbf{x}_i - \theta_t$ is small. More specifically,

$$\|\mathbf{x}_i - \theta_t\|_2 \leq \|\mathbf{x}_l - \theta_t\|_2, \forall i \in S_{\mathrm{bad}} \cap S_t, \forall l \in S^\star\backslash S_t.$$

For any good sample with index $l$, the value $\|\mathbf{x}_l - \theta_t\|_2 \leq \|\mathbf{x}_l - \theta^\star\|_2 + \|\theta^\star - \theta_t\|_2$ by triangle inequality, where the square of $\frac{1}{\sigma}\|\mathbf{x}_l - \theta^\star\|_2$ follows $\chi^2(d)$ distribution. Denote $\mathcal{B}_d(q)$ be the $q$-quantile value of $\chi^2(d)$ distribution, with high probability, the $q$-quantile of $\frac{1}{\sigma}\|\mathbf{x}_l - \theta^\star\|_2$ for all good samples is bounded by $2\sqrt{\mathcal{B}_d(q)}$. Next, we have

$$\theta_{t+1} - \theta^\star = \frac{1}{\tau n}\left(\sum_{i\in S_t\backslash S^\star}\mathbf{x}_i - \sum_{i\in S^\star\backslash S_t}\mathbf{x}_i\right) + (\bar{\theta} - \theta^\star)$$

which is based on the definition in (13) and (14). Then,

$$\|\theta_{t+1} - \theta^\star\|_2$$

$$= \frac{1}{\tau n}\left\|\sum_{i\in S_t\backslash S^\star}(\mathbf{x}_i - \theta_t + \theta_t) - \sum_{i\in S^\star\backslash S_t}(\mathbf{x}_i - \theta^\star + \theta^\star) + (\bar{\theta} - \theta^\star)\right\|_2 \tag{15}$$

$$\leq \frac{1}{\tau n}\left(|S^\star\backslash S_t|\,\|\theta^\star - \theta_t\|_2 + \left\|\sum_{j\in[m]}\sum_{i\in S_{\mathrm{bad}}^j\cap S_t}(\mathbf{x}_i - \theta_t)\right\|_2 + \left\|\sum_{i\in S^\star\backslash S_t}\mathbf{x}_i - \theta^\star\right\|_2\right) + \|\bar{\theta} - \theta^\star\|_2 \tag{16}$$

$$\leq \frac{1}{\tau n}\left(|S^\star\backslash S_t|\,\|\theta^\star - \theta_t\|_2 + |S_t\backslash S^\star|\left(\|\theta_t - \theta^\star\|_2 + 2\sigma\sqrt{\mathcal{B}_d(\frac{|S_t\cap S^\star|}{\tau^\star n})}\right) + \left\|\sum_{i\in S^\star\backslash S_t}\mathbf{x}_i - \theta^\star\right\|_2\right) + \|\bar{\theta} - \theta^\star\|_2 \tag{17}$$

(15) makes an expansion for each term: the norm of $\mathbf{x}_i - \theta_t$ for $i \in S_t\backslash S^\star$ should be small because of the selection rule, while the norm of $\mathbf{x}_i - \theta^\star$ for $i \in S^\star\backslash S_t$ should be small because they are good samples. $\mathcal{B}_d\left(\frac{|S^\star\cap S_t|}{\tau^\star n}\right) = O(d)$. Inequality (16) splits (15) into 4 terms. The second term in (16) is bounded based on triangle inequality and the fact that if a sequence is element-wise larger than the other sequence, then the order statistics is also larger. For the second last term in (17), since the set $S^\star$ is selected by spanning a ball centered at $\theta^\star$, the distance to the center from any sample in set $S^\star$ is bounded by $c\sigma\sqrt{d}$ for some constant $c$. Therefore,

$$\|\theta_{t+1} - \theta^\star\|_2 \leq \frac{2|S^\star\backslash S_t|}{\tau n}\|\theta^\star - \theta_t\|_2 + \frac{c|S^\star\backslash S_t|}{\tau n}\sigma\sqrt{d} + c\frac{\log n}{\sqrt{\tau n}}\sigma\sqrt{d}. \tag{18}$$

In order for (18) to converge, we expect $|S^\star\backslash S_t| < \tau n/2$. For $\tau^\star \geq \frac{1}{2}$, given $\|\theta_t - \theta^\star\|_2 \leq 0.9\min_{j\in[m]}\|\theta_t - \theta^j\|_2$, there exists $c_1 \in (0,1)$, such that $|S^\star\backslash S_t| \leq \frac{c_1}{2}\tau n$ with high probability (i.e., more than half of samples in $S_t$ are from the correct component). As a consequence,

$$\|\theta_{t+1} - \theta^\star\| \leq c_1\|\theta_t - \theta^\star\| + c_2\sigma\sqrt{d}. \tag{19}$$

For $\tau < \frac{1}{2}$, in order to guarantee $|S^\star\backslash S_t| < \tau n/2$, we not only require $\|\theta_t - \theta^\star\|_2 \leq 0.9\min_{j\in[m]}\|\theta_t - \theta^j\|_2$. This is because, the samples generated from $\theta^\star$ may be a small fraction of the whole dataset (less than half), and when $\theta^t$ is slighly closer to $\theta^\star$, it will still collect more bad samples because of the higher density. Intuitively, when $\sigma$ is large enough, for any $\theta_t \in \mathbb{R}^d$, $|S^\star\backslash S_t|$ can be larger than $\tau n/2$. Observe that the median of squared distance from the good samples to $\theta_t$ (denoted as $\mathtt{med}_{\mathrm{good}}$) concentrates at the sample mean, which concentrates at the expected mean, which is $\|\theta_t - \theta^\star\|_2^2 + \sigma^2 d$. On the other hand, we require the number of bad samples whose squared distance to $\theta_t$ is less than $\mathtt{med}_{\mathrm{good}}$ is less than $\tau n/2$, which corresponds to the $\frac{\tau}{2(1-\tau)}$-quantile value to be greater than $\mathtt{med}_{\mathrm{good}}$. Therefore, a valid $\theta_t$ should satisfy:

$$\mathcal{P}\left[\sigma^2\mathbf{n}^\top\mathbf{n} + 2\sigma\mathbf{n}^\top(\theta_t - \theta^{(j)}) < \|\theta_t - \theta^\star\|_2^2 - \|\theta_t - \theta^{(j)}\|_2^2 + \sigma^2 d\right] \leq \frac{\tau}{2(1-\tau)},$$

where $\mathbf{n} \sim \mathcal{N}(0, I_d)$ is a normal Gaussian vector. By sub-exponential inequality, and reorganizing the terms, a sufficient condition we require is $\|\theta_t - \theta^\star\|_2 \leq \min_{j\in[m]}\|\theta_t - \theta^j\|_2 - c\sqrt{\log\frac{2(1-\tau)}{\tau}}\sigma$. This condition essentially implies a $\sqrt{\log k}\sigma$ separation distance, which is the lower bound for computationally efficient algorithm to learn a Gaussian mixture model according to Regev & Vijayaraghavan (2017).

## D.2 SIMULATION RESULTS

In this part, we provide synthetic experimental results for the Gaussian mixture model. Most of the settings are similar to the settings for mixed linear regression in Section C.2. For clearness, we will re-state some of the definitions/settings.

### D.2.1 ILFB ON GAUSSIAN MIXTURE MODEL

We verify the performance of ILFB for Gaussian mixture model through synthetic experiments. In Gaussian mean estimation, we are interested in recovering the mean of the dominant mixture component. The bad samples are generated from another Gaussian distribution, and the distance of the two centers is set as 1 for convenience. We test the performance when $\tau^\star$ varies in the interval $[0.5, 0.8]$ with $\sigma = 0.15$. We compare:

(1) *MLE*: All samples are treated as good samples to find the best parameter;

(2) *Oracle*: learn from the $\tau^\star n$ good samples given oracle access;

(3) *Super-Oracle*: learn from $n$ samples given oracle access;

(4) *R-Init+ILFB* : ILFB with random initialization;

(5) *M-Init+ILFB* : ILFB using *MLE* as initialization.

According to Fig. 9 (left), ILFB performs close to the oracle, and is much better than the naive method and ILFB with random initialization. In Fig. 9 (right), we see linear convergence at the first few rounds until reaching a static point.

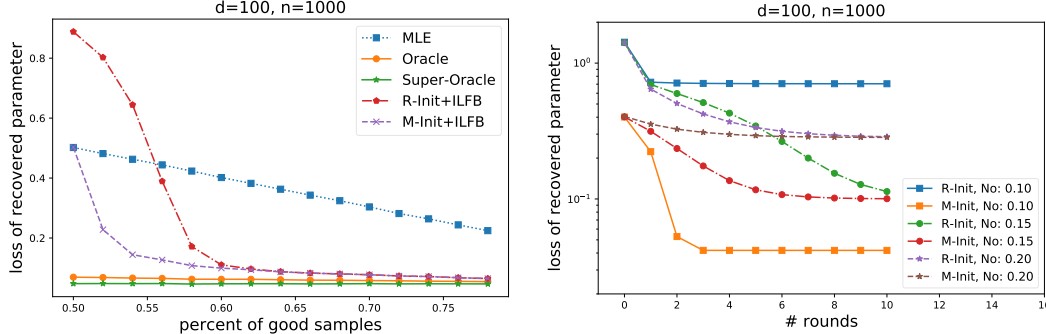

Figure 9: **Left:** Performance of different algorithms for Gaussian mixture model, measured by the $\ell_2$ loss of the recovered parameter to the true parameter, $\sigma = 0.15$. **Right:** Convergence speed of ILFB under different noise levels, using both MLE initialization and random initialization.

### D.2.2 ASYMPTOTIC PERFORMANCE

Next, we check the asymptotic behavior of ILFB for Gaussian mixture model under different noise levels. As we stated in our theorem, ILFB will converge to the noise ball centered at the true parameter, which does not guarantee consistency. In fact, our experiments verifies that the result is tight.

Our experimental setting is as follows: we consider mixture of two Gaussian distributions, where $60\%$ of the data comes from the interested component. The distance between two centers is 1. We test the performance under different sample sizes, varies from 100 to 51200. The standard deviation of each result is calculated based on 100 random runs.

In Fig. 10 (middle), there exists a gap between ILFB and the oracle, i.e., the oracle goes asymptotically to the true parameter as $n$ gets larger, while the performance of ILFB does not decrease with $n$ after some threshold. The same problem exists when the noise becomes larger, as shown in Fig. 10, however, for $n \leq 51200$, the results for both ILFB and the oracle are dominated by the noise. In the small noise regime, both ILFB and the oracle perform well.

We also want to point out that this pheonema of non-exact recovery when the noise is large is not a problem introduced by our algorithm. In fact, for a series of work on learning Gaussian mixture models (estimating the means given identity covariance matrix), a *separation* between every pair of mixture components is required in order to learn in polynomial time (notice that this polynomial time is usually much larger compared to our algorithm which is near-linear). For Fig. 10 (middle and right), this separation requirement is clearly not satisfied.

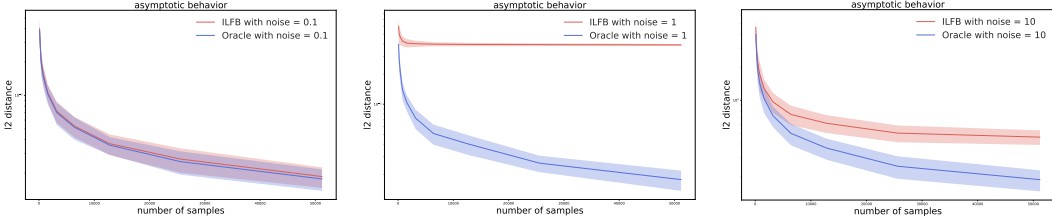

Figure 10: Performance of ILFB for Gaussian mixture model under different noise levels ($\sigma = 0.1, 1, 10$ from left to right). For small noise (left plot), the performance of ILFB decreases along with the oracle performance. When the noise gets larger (middle plot), we observe a gap between ILFB and the oracle when sample size $n$ increases. When the noise gets dominate the signal (right plot), the performance of ILFB and the oracle are not far apart for $n \leq 51200$, but the gap will get larger asymptotically.

### D.3 INITIALIZATION FOR GAUSSIAN MIXTURE MODEL

We present a performance guarantee of the simple initialization step, which takes the average over all the samples. More specifically, we show that the initialization will be closer to the dominant component under mild assumptions.

**Theorem 7 (initialization).** *Given a dataset $\mathcal{D}$ generated following (2), for a certain $j$, define $d_m^j := \min_{l \in [m] \setminus \{j\}} \|\theta^{(j)} - \theta^{(l)}\|_2$. Consider taking the overall mean as the initialization $\theta_0$, then, if $\left\| \sum_{l \in [m] \setminus \{j\}} (\theta^{(j)} - \theta^{(l)}) |S^l| \right\|_2 \leq \alpha n d_m^j$, and $n \geq \frac{cd\sigma^2}{(\frac{9}{19} - \alpha)^2 (d_m^j)^2}$, we have $\|\theta_0 - \theta^{(j)}\|_2 \leq 0.9 \min_{l \in [m] \setminus \{j\}} \left\| \theta_0 - \theta^{(l)} \right\|_2$ with high probability.*

*Proof.* Assume $\left\| \sum_{j \in [m]} (\theta^\star - \theta^{(j)}) |S_{\text{bad}}^j| \right\|_2 \leq \alpha n \min_{j \in [m]} \left\| \theta^\star - \theta^{(j)} \right\|_2$. Consider the initialization

$$\theta_0 = \frac{1}{n} \sum_{i=1}^n \mathbf{x}_i.$$

We have with high probability,

$$
\begin{aligned}
\|\theta_0 - \theta^\star\|_2 &= \frac{1}{n} \left\| \sum_{i=1}^n \mathbf{x}_i - \theta^\star \right\|_2 = \frac{1}{n} \left\| \left[ \sum_{i \in S_{\text{good}}} \mathbf{x}_i - \theta^\star + \sum_{i \in S_{\text{bad}}} \mathbf{x}_i - \theta^\star \right] \right\|_2 \\
&\leq \frac{1}{n} \left\| \sum_{j \in [m]} (\theta^\star - \theta^{(j)}) |S_{\text{bad}}^j| \right\|_2 + c\sigma \sqrt{\frac{d}{n}} \\
&\leq \frac{1}{n} \alpha n \min_{j \in [m]} \left\| \theta^\star - \theta^{(j)} \right\|_2 + c\sigma \sqrt{\frac{d}{n}} \\
&\leq \frac{9}{19} \min_{j \in [m]} \left\| \theta^\star - \theta^{(j)} \right\|_2
\end{aligned}
$$

if $n \geq \frac{cd\sigma^2}{(\frac{9}{19} - \alpha)^2 d_k^2}$. This will imply $\|\theta_0 - \theta^\star\|_2 \leq 0.9 \min_{j \in [m]} \left\| \theta_0 - \theta^{(j)} \right\|_2$. $\blacksquare$

# E    MORE EXPERIMENTAL RESULTS

Table 4: MNIST classification: comparison with other choices

| dataset | MNIST | | | | | | |
|---|---|---|---|---|---|---|---|
| $\tau^\star = \frac{\# \text{ good}}{\# \text{ total}}$ | **Baseline** | **ILFB** | **Centroid** | **1-Step** | $\Delta\tau = 10\%$ | $\Delta\tau = 15\%$ | $\Delta\tau = 20\%$ |
| 60% | 70.26 | 90.00 | 74.52 | 78.48 | 88.24 | 86.48 | 84.49 |
| 70% | 85.95 | 92.09 | 88.47 | 89.31 | 90.83 | 89.19 | 88.10 |
| 80% | 92.62 | 94.30 | 92.79 | 92.78 | 92.78 | 90.17 | 88.63 |
| 90% | 93.88 | 94.41 | 94.32 | 93.94 | 92.88 | 91.21 | 89.90 |

Table 5: MNIST GAN: comparison with other choices

| dataset | MNIST | | | | | | |
|---|---|---|---|---|---|---|---|
| $\tau^\star = \frac{\# \text{ good}}{\# \text{ total}}$ | **Baseline** | **ILFB** | **Centroid** | **1-Step** | $\Delta\tau = 10\%$ | $\Delta\tau = 15\%$ | $\Delta\tau = 20\%$ |
| 70% | 70 | 97.00 | 61.46 | 77.77 | 83.33 | 78.06 | 83.59 |
| 80% | 80 | 100.00 | 77.46 | 76.84 | 98.80 | 99.56 | 97.77 |
| 90% | 90 | 100.00 | 89.57 | 91.90 | 98.85 | 99.01 | 98.04 |

Table 6: MNIST classification with random error.

| dataset | MNIST | | | | | | |
|---|---|---|---|---|---|---|---|
| $\tau^\star = \frac{\# \text{ good}}{\# \text{ total}}$ | **Baseline** | **ILFB** | **Centroid** | **1-Step** | $\Delta\tau = 10\%$ | $\Delta\tau = 15\%$ | $\Delta\tau = 20\%$ |
| 30% | 82.74 | 87.88 | 82.72 | 85.17 | 85.30 | 75.14 | 61.52 |
| 50% | 90.57 | 91.80 | 91.00 | 91.48 | 90.43 | 87.37 | 78.21 |
| 70% | 93.04 | 92.40 | 93.47 | 92.55 | 91.14 | 89.39 | 87.90 |
| 90% | 93.98 | 93.93 | 94.14 | 93.58 | 92.32 | 90.61 | 89.95 |

**Experimental settings:**    In this section, we present additional experimental results, in order to verify the performance of ILFB under different parameter settings, and compare with other algorithms. More specifically, we present the results using the following methods/algorithms:

- **Baseline**: naive trainig using all the samples;
- **ILFB** : our proposed iterative learning algorithm with 5 iterations, using a mis-specified $\tau$ which is 5% less than the true value;
- **Centroid**: using the centroid of the input data to filter out outliers. For classification task, we calculate the centroids for the samples with the same label/class and filter each class separately;
- **1-Step**: **ILFB** algorithm with a single iteration;
- $\Delta\tau = \tau^\star - \tau \in \{10\%, 15\%, 20\%\}$: **ILFB** under different mis-specified $\tau$ value,

under the following three problem settings:

- Classification with *systematic error*;
- MNIST generation with Fashion-MNIST images ;
- Classification with *random label error*.

For classification tasks (Table 4 and Table 6), we show the best accuracy on the validation set. For the generation task (Table 5), we present the ratio of true MNIST samples selected by each method. For the baseline method, since the DC-GAN is trained using all samples, the reported value is exactly the $\tau^\star$.

**Results:**  Table 4 shows the performance under systematic error, for $\tau^\star$ varies from $60\%$ to $90\%$. ILFB not only performs better than the baseline, but also outperform the centroid method by a large margin, especially when the dataset is noisy ($\tau^\star$ is small). By comparing ILFB with its 1-step counterpart, we see the benefit brought by doing the iterative updating is also significant. Table 5 shows the performance of generation quality under different noise levels. We observe that centroid method does not work, which may due to the fact that all MNIST and Fashion-MNIST images are hard to be distinguished as two clusters in the pixel space. Notice that there are in fact 20 clusters (10 from MNIST, and 10 from Fashion-MNIST), and we are interested in 10 of them. ILFB works well since it automatically learns a clustering rule when generating on the noisy dataset. For example, for $\tau^\star = 80\%$, even with a mis-specified $\tau = 60\%$, ILFB is capable of ignoring almost all bad samples. Again, we also observe significant improvement of ILFB over its 1-step counterpart. In Table 6, we show the performance of ILFB in the random error setting. Again, ILFB performs much better in the extremely noisy setting.

We also have results showing that ILFB works well for generation when the corrupted samples are pure Gaussian noise. However, we do not think it is a practical assumption, and the result is not presented here.

