# OpenReview forum: "Iteratively Learning from the Best"
_ICLR.cc/2019/Conference_

### Official Review · AnonReviewer2 · 2018-10-30
**novelty over early work? different angle to view this problem - sparse learning. Proofs/statements need to more rigorous.**

**Rating:** 6
**Confidence:** 4

**Review:**

This paper provides an algorithm that excludes the bad training data in the training process and obtain a more accurate model for both supervised and unsupervised learning problem. The paper gives the theoretical guarantee for mixed linear regression and Gaussian mixture model, and also conducts the experiments for deep image classification and deep generative models.

Major Concerns:
1, As said in related work, a soft version of this paper’s method has been proposed in the previous work, and the major seems to be that there is no initialization in the previous work which only leads to local convergence. Therefore, based on my understanding, the only innovation in this paper is that it gives the initialization process so that the algorithm can converge to the global optimal solution. But even this innovation only successes on some specific problems (Section 4-7). There are too few innovations.

2, In Section 4, for mixed linear regression, Theorem 1 and Theorem 2 together can not guarantee the global optimal solution for the algorithm. The author should demonstrate  “strict inequality” property in the 3rd line in Theorem 2, because it should correspond to the  “strict inequality” property in the 2nd line in Theorem 1.

3. Another angle to view the target problem in paper is from the outlier detection problem. The sparse learning formulation and theory can be conducted to solve this problem. Many existing theoretical analysis methods and optimization methods can be applied. For example, authors can refer to

A Robust AUC Maximization Framework With Simultaneous Outlier Detection and Feature Selection for Positive-Unlabeled Classification, 2017

The comparison to these type of methods need to be included.

Minor Concerns:
1, Theorem 2 does not give the probability, only mentioning “high probability”. How high? I do not find the probability in the proof as well. The same concern happens to Theorem 4. I think that

2, In Section 6 and 7, the author does not compare with other algorithms, which can not show the advantage of this algorithm.

---

> ### Author Response · Authors · 2018-11-23
> **Thank you for your feedback**
>
> We thank the reviewer for the feedback. Regarding your concerns, we want to make several clarifications:
>
> Review: "As said in related work, a soft version of this paper’s method has been proposed in the previous work, and the major seems to be that there is no initialization in the previous work which only leads to local convergence. Therefore, based on my understanding, the only innovation in this paper is that it gives the initialization process so that the algorithm can converge to the global optimal solution. But even this innovation only successes on some specific problems (Section 4-7). There are too few innovations."
>
> Response: The main innovations of our paper over the existing work is:
> 1.    For GMMs and Mixed Linear Regression, we provide strong statistical consistency guarantees, while the previous work just provided loss-function minimization and generalization, or robustness for only single linear regression and very small fractions. In fact for Mixed Linear Regression our results provide the best guarantees, even when compared to methods specially made for that problem (while our method is just an instance of a more general idea)
> 2.    We provide strong empirical evidence that the idea of retaining the best samples has merit for deen neural classifiers and generative models, something not shown in previous work.
> 3.    The previous work on the soft version had one more penalty parameter (needed to prevent the “soft” from becoming “hard”), and choosing that parameter is not easy, while still being important to success - even if one knows the fraction of bd samples. We do not have such a parameter problem.
>
> Review: "In Section 4, for mixed linear regression, Theorem 1 and Theorem 2 together can not guarantee the global optimal solution for the algorithm. The author should demonstrate  “strict inequality” property in the 3rd line in Theorem 2, because it should correspond to the  “strict inequality” property in the 2nd line in Theorem 1."
>
> Response: We have updated the theorems of the paper (please see the new version), and these do not have this issue.
>
> Review: "Another angle to view the target problem in paper is from the outlier detection problem. The sparse learning formulation and theory can be conducted to solve this problem. Many existing theoretical analysis methods and optimization methods can be applied. For example, authors can refer to
> A Robust AUC Maximization Framework With Simultaneous Outlier Detection and Feature Selection for Positive-Unlabeled Classification, 2017
> The comparison to these type of methods need to be included. "
>
> Response: We thank the reviewers for bringing this relevant work to our attention. We have included this in the related work section of the updated version.
>
> Review: "Theorem 2 does not give the probability, only mentioning “high probability”. How high? I do not find the probability in the proof as well. The same concern happens to Theorem 4. I think that"
>
> Response: By high probability, we mean with probability 1-n^{-c} for some constant c, which is consistent for all the theorems. We have made this explicit in the updated version.
>
> Review: "In Section 6 and 7, the author does not compare with other algorithms, which can not show the advantage of this algorithm."
>
> Response: In our revised version, in Appendix E, we provide a comparison with the 'distance to centroid' method to give an idea of ILFB's advantage over other simple algorithms. We also show (in Appendix E) the significant improvement over ILFB's 1-step counterpart, which is a natural algorithm to deal with noisy data. We observe empirically that a batch version of our algorithm (gradient descent over a subset of a batch of samples with smaller loss) is unstable, and would stuck at bad local results. More specifically, in classification task, the batch version improves over the baseline, but is worse than our current algorithm. In the generation task, the batch version will get stuck. We have not included the results for the batch version, since it may be due to the reason that we did not select hyperparameters correctly (maybe a much smaller learning rate would work). On the other hand, our current algorithm does not need any hyperparameter adjustment if one knows how to train the network in the standard, noiseless setting.

---

### Official Review · AnonReviewer1 · 2018-11-02
**A well written paper but has major problems**

**Rating:** 3
**Confidence:** 5

**Review:**

This paper introduces a framework for situation when the training samples are not pure. The idea is a simple approach by training a model and removing a portion of examples from the training set based on the loss of the model. The authors provide some theoretical study on two models: linear regression and Gaussian mixture model and utilize deep neural network to show their framework performs well experimentally.

My main problem with this work is the difficulty in understanding whether the reason our training model produces a large loss on some examples is due to them being bad examples or is because the model is not good enough and needs improvement. For example, one can always overtrain a classifier such that it classifies the training examples perfectly. Now the question become how much should I train my classifier. In case of Deep Neural Networks for example, the number of epochs can change the loss occurred by classifiers on the examples and it is not easy to know when to stop training in order to utilize the procedure introduced in this work.


The theoretical work is related to linear regression and Gaussian Mixture model but the experiments are relayed to Deep Neural Nets! So I am not sure if this setup makes sense. Either both should be for DNN or neither should be.

I am not sure if I understand Section 5 and the discussion related to the Gaussian Mixture Model. In Gaussian Mixture model, there are multiple components and each commonest has its own parameteres. So not sure (1) why the authors assume only mean parameter. (2) Given that Gaussian mixture model assumes multiple components, doesn't it automatically address the problem by putting the samples from different distribution in a different component?

Page 5 typo: closest point closest to

The parameter \tau is set to 5 percent less that the true ratio of good samples (correct labels). This seems a pretty bias choice and implicitly applied that one needs to know the true value of this ratio which is a huge expectation. The authors need to investigate the effect of the changes of this value on the performance of their proposed framework! To me, it seems that the results can be hugely affected by the value of this parameter.

The experiment with GAN is very wired. How can you expect to have a data set with 20 percent of its examples be bad cases. The authors need to justify that such cased can happen in real applications.

---

> ### Author Response · Authors · 2018-11-23
> **Thank you for your feedback**
>
> We thank the reviewer for the feedback. Regarding your concerns, we want to make several clarifications:
>
> Review: “My main problem with this work is the difficulty in understanding whether the reason our training model produces a large loss on some examples is due to them being bad examples or is because the model is not good enough and needs improvement. “
>
> Response: Our work does not consider the problem of how much to train / avoid overtraining a deep classifier; this is of course an important issue but is more broader and orthogonal to our focus. We use the standard technique of (a) running a fixed maximum number of epochs, and then (b) choosing the model of these that has the best validation set performance. Indeed our objective is to show that sample rejection (using our method) can work with existing training choices.
>
> Review: "The theoretical work is related to linear regression and Gaussian Mixture model but the experiments are relayed to Deep Neural Nets! So I am not sure if this setup makes sense. Either both should be for DNN or neither should be."
>
> Response: We do have experiments for the Gaussian Mixture and Mixed Linear Regression models as well, in the appendix; they match the theory results presented in terms of showing strong statistical consistency. It would indeed be great to get similar strong statistical consistency results for Deep Neural Nets; however, these do not exist even for the standard case when there is no bad data (i.e. even given samples generated according to a DNN, it is not known if the DNN parameters can be consistently found by standard SGD-based training). Rather, our DNN experiments show that the basic IDEA - rejecting high-loss samples - has (empirical) value even in DNN settings.
>
> Review: "In Gaussian Mixture model, there are multiple components and each commonest has its own parameteres. So not sure (1) why the authors assume only mean parameter. (2) Given that Gaussian mixture model assumes multiple components, doesn't it automatically address the problem by putting the samples from different distribution in a different component?"
>
> Response: For point (1) above, Our GMM analysis assumes the means of each component are the only unknown parameters, the covariances are known to be \sigma I for each component. This is a standard analytical setting for GMMs, see e.g. [RV17] and many references therein.
> For point (2) above, indeed different components means different distributions. Note however that it may still not be easy to distinguish them in an unlabeled dataset.
> Overall: our algorithm for the GMM can be best understood as a “1-means” algorithm, where we try to find the SINGLE gaussian that best fits a subset of the samples, given a dataset of samples coming from multiple overlapping gaussians. Our theorem shows that such a 1-means algorithm succeeds.
>
> [RV17] On Learning Mixtures of Well-Separated Gaussians. Oded Regev, Aravindan Vijayaraghavan. FOCS 2017
>
> Review: "The parameter \tau is set to 5 percent less that the true ratio of good samples (correct labels). This seems a pretty bias choice and implicitly applied that one needs to know the true value of this ratio which is a huge expectation. The authors need to investigate the effect of the changes of this value on the performance of their proposed framework!"
>
> Response: Indeed we do need to know an upper bound on what fraction of data that is corrupted (but obviously not which precise samples are corrupted). We note that even with this knowledge there do not exist methods that can effectively learn good models. We do not need to know the fraction exactly.
> That said, we did do preliminary investigations of what happens if our assumed fraction is incorrect - i.e. we retain more samples in each step than is warranted by the data. Empirically, it seems the quality of the model gently degrades with this over-estimation, while still being better than the baseline standard practice of just blindly using all data. In our revised version, we have included more simulations (Appendix A) and experiments (Appendix E) to show that our algorithm is not very sensitive to mis-specified tau.
>
> Review: "The experiment with GAN is very wired. How can you expect to have a data set with 20 percent of its examples be bad cases. The authors need to justify that such cased can happen in real applications. "
>
> Response: Indeed if a lot of quality human effort has been spent in developing a curated dataset, there will not be significant fractions of errors. However the motivation for our work is that this effort is expensive, and sloppy data curation can be a reality if this expense is not undertaken. Say for example one wants to make a GAN for impressionist paintings; ideally one would need a clean dataset of only such paintings - curated from a larger one of all kinds of paintings. Bad curation can result in significant fraction of samples being non-impressionist. Similar examples are easy to imagine in natural language etc.

---

### Official Review · AnonReviewer3 · 2018-11-11
**Tackles important problem but needs more fleshing out**

**Rating:** 6
**Confidence:** 3

**Review:**

The authors propose an iterative method for discarding outlying training data: first, learn a model on the entire training dataset; second, identify the training examples that have high loss under the learned model; and then alternate between re-learning the model on the training examples that do not have high loss, and re-identifying the training examples with high loss under the new model. This method works for both supervised and unsupervised learning, and the authors show that in theory, their method has some convergence properties in the mixed linear regression and Gaussian mixture model settings. The authors also run some experiments on neural networks and datasets with synthetic noise to show the benefits of their proposed method.

The problem of noisy datasets is relevant to almost all machine learning problems in the real world, and the authors' method shows promise as a straightforward way to increase performance on such noisy data. However, my opinion is that the authors need to make a stronger case, theoretically and/or experimentally, for why their method should be preferred to other methods. Detailed comments follow.

== Experiments ==
1) No comparisons are provided to other outlier detector methods (e.g., based on nearest neighbors, distance to centroid, influence functions, etc.) or techniques that also purport to deal with noisy labels (e.g., by modifying the learning algorithm or loss function). While there are too many existing methods to expect the authors to benchmark against all of them, it's important to at least have a couple of representative comparisons.

2) It'd be nice to have an ablative analysis to tease out the factors behind the gain in accuracy. For example, is the iteration important, or would a single pass suffice? How robust is the algorithm to tau, the fraction of data to discard? (The authors do test initializing randomly vs. initializing on the full dataset.)

3) The systematic label noise scenario seems to favor the authors' method (though the authors claim that it is a harder scenario than random label noise). It'd be helpful to see if the method works against random noise.

== Theory ==
4) The assumptions seem very restrictive. For example, for mixed linear regression (section 4), the features of all examples are assumed to be drawn i.i.d. from an isotropic Gaussian (so even the bad samples are drawn from the same distribution as the good samples; and all features are independent). To my knowledge, this is not a "standard and widely studied" assumption. For the Gaussian mixture model (section 5), a similar isotropic Gaussian assumption is made for each mixture.

5) Beyond the independence assumptions mentioned above, the initialization results make additional assumptions on the "bad" data (e.g., average distance of the good vs. bad parameters) that I found hard to parse. How strong are these assumptions? Do they hold on real datasets?

6) The convergence results (Theorems 1 and 3) have a constant term sigma in them. This is surprising and seems to me to considerably weaken the result -- one would expect that the dependence on sigma will decrease with n.

I think either a strong experimental or strong theoretical section would be sufficient for me to recommend acceptance. However, the paper currently shows potentially interesting experimental/theoretical results but does not do a comprehensive job of either side.

---

> ### Author Response · Authors · 2018-11-23
> **Thank you for your feedback**
>
> We thank the reviewer for the feedback. Regarding your concerns, we want to make several clarifications:
>
> Review: "1) No comparisons are provided to other outlier detector methods or techniques that also purport to deal with noisy labels. While there are too many existing methods to expect the authors to benchmark against all of them, it's important to at least have a couple of representative comparisons.
> 2) It'd be nice to have an ablative analysis to tease out the factors behind the gain in accuracy. For example, is the iteration important, or would a single pass suffice? How robust is the algorithm to tau, the fraction of data to discard?"
>
> Response: In our revised version, we include a more holistic comparison in Appendix E. First, we include a comparison with the centroid method. Second, according to the reviewer's suggestion, we have an ablation study, and show the benefit of doing the iterative update compared to a single pass, and the robustness of the algorithm to tau. In fact, our original experiments with tau being 5% less than the true value is not hard to be satisfied in practice, since in the worst case, one can sweep over all possible tau s with an increment of 5% each time.  In Appendix A (Figure 5), we also include a simulation showing the robustness to tau in the linear regression setting.
>
> Review: "3) The systematic label noise scenario seems to favor the authors' method (though the authors claim that it is a harder scenario than random label noise). It'd be helpful to see if the method works against random noise."
>
> Response: In Appendix E (Table 6), we add new experiments showing our algorithm works decently. In the random label noise setting, the contribution of the bad samples are 'canceled out' naturally due to the randomness assumption. It is empirically observed (see [1]) in this setting that when the sample size is large enough, even if we train with the naive method, we would still get pretty good accuracy.
> [1] Deep Learning is Robust to Massive Label Noise. David Rolnick, Andreas Veit, Serge Belongie, Nir Shavit
>
> Review: "4) The assumptions seem very restrictive. For example, for mixed linear regression (section 4), the features of all examples are assumed to be drawn i.i.d. from an isotropic Gaussian. To my knowledge, this is not a "standard and widely studied" assumption. For the Gaussian mixture model (section 5), a similar isotropic Gaussian assumption is made for each mixture."
>
> Response: Our assumptions are indeed standard:
> The following GMM papers make exactly the same assumptions:
> [2] On Learning Mixtures of Well-Separated Gaussians. Oded Regev, Aravindan Vijayaraghavan. FOCS 2017
> [3] Global Analysis of Expectation Maximization for Mixtures of Two Gaussians. Ji Xu, Daniel Hsu, and Arian Maleki. NIPS 2016
>
> The following Mixed Linear Regression papers make stronger assumptions where they consider the noiseless setting:
> [4] Xinyang Yi, Constantine Caramanis, and Sujay Sanghavi. Alternating minimization for mixed linear regression. In ICML 2014
> [5] Kai Zhong, Prateek Jain and Inderjit S. Dhillon.  Mixed Linear Regression with Multiple Components. NIPS 2016
>
> Only very recently the following paper considers the setting where the covariance for each component needs not be identity.
> [6] Yuanzhi Li and Yingyu Liang. Learning mixtures of linear regressions with nearly optimal complexity. COLT 2018
> However, (1) they consider the noiseless case, (2) their runtime is exponential in the number of mixture components, even for recovering a single component.
>
> Review: "5) Beyond the independence assumptions mentioned above, the initialization results make additional assumptions on the "bad" data (e.g., average distance of the good vs. bad parameters) that I found hard to parse. How strong are these assumptions? Do they hold on real datasets?"
>
> Response: In the updated paper we have removed the initialization, since in any case that only guaranteed convergence to the top component. We instead emphasize (both here and in the revised paper) that the local convergence result implies that we will converge to the true component closest to our initial point - this can thus be easily used with multiple different “far away” initializations to get a good estimate for all components.
>
> Review: "6) The convergence results (Theorems 1 & 3) have a constant term sigma in them. This is surprising and seems to me to considerably weaken the result -- one would expect that the dependence on sigma will decrease with n."
>
> Response: For Gaussian mixture model setting, even if we do thresholding starting from the true mean, the other component will contribute a fraction (a function of sigma) of bad samples, which will make the next iteration estimate to be biased within the noise ball. Similarly things happen for mixed linear regression.  We have added this part of discussion to the revised paper. We also experimentally validate the performance under different noise levels in Appendix C.2 and D.2..

---

### Meta-Review · Area_Chair1 · 2018-12-15
**problems with experiments and assumptions; post-deadline revision too large**

**Confidence:** 5
**Recommendation:** Reject

**Metareview:**

This paper addresses the problem of learning with outliers, which many reviewers agree is an important direction. However, reviewers point to issues with the experiments (missing baselines, ablations, etc.) and are concerned that the assumptions in the theoretical analysis are too strong. These were potentially addressed in a revised version of the paper, but the revisions are so major that I do not think it is appropriate to consider them in the review process (and it is hard to assess to what extent they address the issues without asking reviewers to do a thorough re-appraisal, which goes beyond the scope of their duties). I encourage the authors to take reviewer comments into account and prepare a more polished version of the manuscript for future submission.